# ATTRIBUTION-GUIDED DECODING

**Piotr Komorowski**[1]   **Elena Golimblevskaia**[1]   **Reduan Achtibat**[1]   **Thomas Wiegand**[1,2,3]
**Sebastian Lapuschkin**[1,4,†]   **Wojciech Samek**[1,2,3,†]

[1]Department of Artificial Intelligence, Fraunhofer Heinrich Hertz Institute
[2]Department of Electrical Engineering and Computer Science, Technische Universität Berlin
[3]BIFOLD - Berlin Institute for the Foundations of Learning and Data
[4]Centre of eXplainable Artificial Intelligence, Technological University Dublin
[†]corresponding authors: {`wojciech.samek,sebastian.lapuschkin`}@hhi.fraunhofer.de

## ABSTRACT

The capacity of Large Language Models (LLMs) to follow complex instructions and generate factually accurate text is critical for their real-world application. However, standard decoding methods often fail to robustly satisfy these requirements, while existing control techniques frequently degrade general output quality. In this work, we introduce Attribution-Guided Decoding (AGD), an interpretability-based decoding strategy. Instead of directly manipulating model activations, AGD considers a set of high-probability output token candidates and selects the one that exhibits the highest attribution to a user-defined Region of Interest (ROI). This ROI can be flexibly defined over different parts of the model's input or internal components, allowing AGD to steer generation towards various desirable behaviors. We demonstrate AGD's efficacy across three challenging domains. For instruction following, we show that AGD significantly boosts adherence (*e.g.,* improving the overall success rate on Llama 3.1 from 66.0% to 79.1%). For knowledge-intensive tasks, we show that guiding generation towards usage of internal knowledge components or contextual sources can reduce hallucinations and improve factual accuracy in both closed-book and open-book settings. Furthermore, we propose an adaptive, entropy-based variant of AGD that mitigates quality degradation and reduces computational overhead by applying guidance only when the model is uncertain. Our work presents a versatile, more interpretable, and effective method for enhancing the reliability of modern LLMs.

## 1 INTRODUCTION

Large Language Models (LLMs) have emerged as powerful tools capable of generating fluent, coherent and contextually relevant text across numerous applications (Brown et al., 2020; Ouyang et al., 2022; Achiam et al., 2023). Despite their success, their reliability is undermined by critical failures, most notably inconsistent adherence to user instructions (Sun et al., 2023; Zhou et al., 2023a; Zeng et al., 2024) and a tendency to generate non-factual, or *hallucinated* (Wei et al., 2024), information. A key enabler of the former has been instruction tuning, which teaches models to better follow human commands expressed in natural language (Zhou et al., 2023b). However, despite these advances, models still struggle to follow complex constraints, especially in lengthy contexts (Liu et al., 2024) or multi-turn dialogues (Li et al., 2024; Qin et al., 2025) where constraints can drift. These shortcomings are not minor flaws but fundamental barriers to deploying LLMs in high-stakes environments that demand precision and trustworthiness.

To address these issues, significant research has focused on developing methods to control and guide the LLM generation process. Standard decoding strategies like top-$k$ (Fan et al., 2018) or nucleus sampling (Holtzman et al., 2020) can modulate the randomness of the output but offer little direct control over semantic properties like factuality or instruction adherence. A more direct line of work involves steering model behavior by directly manipulating the model's internal activations to guide it towards a desired style or content (Li et al., 2023a; Rimsky et al., 2024). While often effective at enhancing the targeted attribute, these interventions come with a significant drawback: a frequent

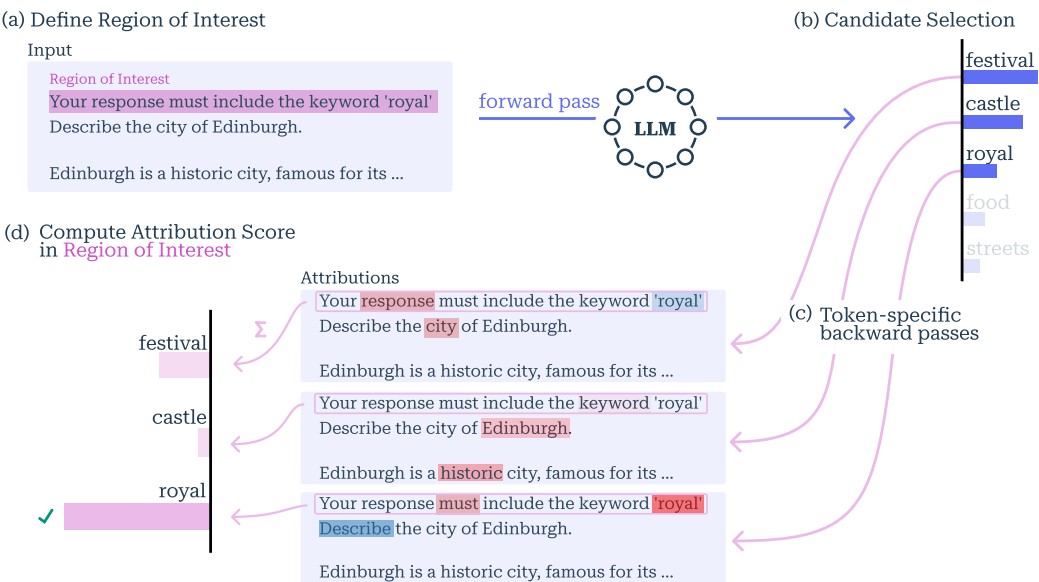

Figure 1: Illustration of the Attribution-Guided Decoding (AGD) framework. **(a)** A Region of Interest (ROI) is defined over a relevant area, such as the user's instruction. **(b)** Next, a standard forward pass generates a candidate set of high-probability tokens, both "festival" (most probable) and "royal" are included. **(c)** An attribution method computes a relevance score for each candidate on the input tokens, quantifying a candidate's dependence on the ROI. **(d)** AGD selects the token ("royal") with the highest aggregated attribution score on the ROI, thereby satisfying the constraint.

degradation of general text quality (Arditi et al., 2024; Stolfo et al., 2025). Altering the internal representations can push the model into out-of-distribution states, leading to increased perplexity, repetitive outputs, and a loss of nuance. This creates an undesirable trade-off where users must choose between better control and higher-quality generation.

In this paper, we ask: can we guide generation towards a desired behavior without directly manipulating the model's internal representations? We propose a new paradigm, Attribution-Guided Decoding (AGD), which reframes decoding as a search for the token that is best justified by a given rationale. The core idea is to leverage post-hoc attribution methods not just for understanding model decisions (Yin & Neubig, 2022), but for guiding them. The process is illustrated in Figure 1. At each generation step, we first identify a small set of plausible next tokens from the model's output distribution. Then, using a feature attribution method, we quantify how much each candidate token *relies* on a specified part of the input, *e.g.* instruction, or model's internals. Finally, the token with the highest attribution score is selected for generation. By restricting its choice to a set of high-probability candidates, AGD maintains fluency and coherence, mitigating degradation in output quality.

AGD is a versatile, fine-tuning-free framework that operates at decoding time, making it broadly applicable to modern LLMs. Although our method incurs additional computational overhead due to the necessity of computing token-level relevance scores, this trade-off results in marked improvements in controllability and interpretability. As contributions, we:

- Introduce AGD, a novel, flexible framework for steering LLM generation via post-hoc analysis of candidate tokens to make generation process more grounded.
- Apply AGD to instruction following and propose an entropy-based adaptive mechanism that dynamically applies guidance, achieving strong instruction adherence while preserving output quality and reducing computational cost.
- Demonstrate the versatility of AGD by targeting parametric knowledge heads to improve factuality and contextual sources to enhance in-context grounding.
- Show that AGD provides insights into interpretability, for instance offering an explanation why certain tokens are chosen over others during the generation process.

## 2 RELATED WORK

**Controlled Text Generation**    Significant research has focused on steering LLM behavior at inference time without costly retraining. One prominent line of work is activation engineering, where steering vectors are added to the model's residual stream to guide its internal states towards desired concepts or styles (Subramani et al., 2022; Burns et al., 2023; Tigges et al., 2024; Rimsky et al., 2024). While powerful, these methods directly intervene in the model's forward pass, fundamentally altering its computation in a way that can harm general output quality (Arditi et al., 2024; Stolfo et al., 2025). Other techniques modify the output logits, often using contrastive decoding approaches to improve properties like factual accuracy (Li et al., 2023b; Shi et al., 2024; Chuang et al., 2024). These methods can be broadly categorized as *interventionist* as they actively modify the model's internal states or output distributions. In contrast, AGD is a *selectionist* method. It does not alter the model's forward pass or logits. Instead, it utilizes model's original output distribution and uses attribution methods as a way to select the candidate that best aligns with a specified goal.

**Instruction-Following**    The ability of LLMs to follow commands has been significantly advanced by instruction tuning (Ouyang et al., 2022; Wei et al., 2022; Gupta et al., 2022; Longpre et al., 2023; Chung et al., 2024), with a corresponding growth in benchmarks for evaluation under varying levels of complexity and context (Zhou et al., 2023a; Zeng et al., 2024; Qin et al., 2024; Jiang et al., 2024). Current post-training methods aimed at enhancing instruction following often require model- and task-specific preparation, such as profiling (Zhang et al., 2024), training linear probes (Heo et al., 2025), or computing steering vectors (Stolfo et al., 2025). This preparation, combined with the need to tune hyperparameters like steering weights and intervention layers, can limit scalability. In contrast, our approach operates entirely at inference time.

**Attribution Methods**    Attribution methods aim to explain a model's prediction by assigning attribution scores to its inputs or internal components. While attention weights are a natural candidate for analysis, their unreliability as faithful explanations motivates the use of saliency-based methods (Bastings & Filippova, 2020). These techniques range from simpler gradient-based methods such as Input×Gradient (I×G) (Simonyan et al., 2014; Sundararajan et al., 2017; Smilkov et al., 2017) to more robust techniques like Layer-wise Relevance Propagation (LRP) (Bach et al., 2015; Voita et al., 2019; Achtibat et al., 2024). Historically, these attribution methods have been used mostly for post-hoc analysis – to understand and debug a model's behavior after a decision has been made (Lapuschkin et al., 2019; Anders et al., 2022; Pahde et al., 2023; Achtibat et al., 2023). To our knowledge, our work is the first to integrate these analytical tools directly into the decoding loop of LLMs, transforming them from a passive, explanatory role into an active, generative one. By doing so, we not only steer the model's output but also provide a rationale for each selection.

## 3 METHOD

An autoregressive language model $\theta$ receives a sequence of input tokens $x = (x_1, x_2, \ldots, x_n)$, referred to as the *prompt*, and generates an output sequence $y = (y_1, y_2, \ldots)$, one token at a time. Let $\mathcal{V}$ denote the model's vocabulary - the full set of discrete tokens that the model can emit. At each decoding step $t$, the model predicts a probability distribution over $\mathcal{V}$, denoted as $\boldsymbol{p}_\theta(y_t \mid x, y_{<t})$, conditioned on the input $x$ and the previously generated prefix $y_{<t} = (y_1, \ldots, y_{t-1})$.

### 3.1 FEATURE ATTRIBUTION

Attribution methods aim to explain a model's prediction by quantifying the contribution of its input or internal components to a specific output. We define $\Omega$ as the set of all attributable components in the model, such as its input token embeddings or attention heads. A general attribution function $\mathcal{A}$ maps a token $c$ to a set of relevance scores over these components:

$$\mathcal{A}_\theta(c \mid x, y_{<t}) \rightarrow \{r_\omega \mid \omega \in \Omega\}, \quad (1)$$

where $r_\omega$ represents the relevance of component $\omega$ to the model's logit for token $c$. In principle, any attribution method could be used, but they involve different trade-offs between faithfulness and computational cost. Perturbation-based methods, while often faithful, are too slow for decoding as they require numerous forward passes (Lundberg & Lee, 2017). Gradient-based methods like I×G

(Simonyan et al., 2014) are more efficient, requiring only a single backward pass, but can produce noisy and unreliable attributions due to the non-linearities in network architectures (Ali et al., 2022).

To balance these factors, we adopt Layer-wise Relevance Propagation (LRP) (Bach et al., 2015), which propagates the output logit value backward through the network in a layer-wise manner. Its adaptation for Transformers, AttnLRP (Achtibat et al., 2024), includes specific rules to handle non-linear components like self-attention and layer normalization, resulting in more stable and faithful relevance scores than simpler gradient methods (Arras et al., 2025). Crucially, LRP is as efficient as gradient-based methods, making it suitable for integration into the decoding loop. Therefore, we select LRP as our primary attribution method, additionally reporting results for I×G for comparison.

## 3.2 ATTRIBUTION-GUIDED DECODING

**Candidate Set Selection**  To control generation, decoding algorithms often restrict sampling to a subset of likely candidates. We restrict sampling to a small but plausible subset of the vocabulary, which we term the *candidate set* $\mathcal{C}_t \subseteq \mathcal{V}$. At each timestep $t$, this set is formed by first selecting the $k$ tokens with the highest probabilities from the distribution $\boldsymbol{p}_\theta(y_t \mid x, y_{<t})$. From this initial set, we then filter out any token whose probability is below a minimum threshold $\pi_{\min}$. This step ensures that we only consider tokens that the model already deems likely, thereby preserving fluency.

**Attribution Scoring**  For each candidate token $c \in \mathcal{C}_t$, we compute an attribution score that quantifies its reliance on a specific Region of Interest (ROI) $R$. It can be defined over any part of the model's input or internal components, such as a subset of input embeddings or specific attention heads, making AGD adaptable to various tasks. The process starts from the model's pre-softmax logit for the candidate token $c$. Using an attribution method $\mathcal{A}$ (e.g., LRP), we backpropagate a signal from this logit to assign relevance scores $r_\omega = \mathcal{A}_\theta(c \mid x, y_{<t}; \omega)$ to the components $\omega$ of the model. The total attribution score $S(c, R)$ for a candidate $c$ with respect to $R \subseteq \Omega$ is the sum of attributions over all components within that region:

$$S(c, R) = \sum_{\omega \in R} r_\omega = \sum_{\omega \in R} \mathcal{A}_\theta(c \mid x, y_{<t}; \omega). \tag{2}$$

A higher score $S(c, R)$ indicates that token $c$ was more influenced by the components in $R$.

**Token Selection**  Finally, we select the token $y_t$ from the candidate set $C_t$ that maximizes the attribution score with respect to the Region of Interest $R$:

$$y_t = \arg\max_{c \in C_t} S(c, R). \tag{3}$$

By replacing the standard probability-maximization objective with an attribution-maximization one, we guide the model to generate tokens that are most consistent with the function encapsulated by $R$.

## 3.3 DEFINING REGION OF INTEREST

The flexibility of AGD lies in how the ROI is defined. By selecting different subsets of the model's attributable components ($R \subseteq \Omega$), we can steer generation towards various desirable behaviors.

**Instruction Following**  For tasks requiring adherence to specific constraints, we partition the input prompt $x$ into an instruction part, $x_I$, and a task-specific query, $x_T$ (example in Table 2). Our objective is to select tokens that are maximally influenced by $x_I$. In this case, the ROI is defined as the set of input token embeddings corresponding to the instruction part of the prompt:

$$R_I = \{e_i \mid x_i \in x_I\}, \tag{4}$$

where $\{e_1, \ldots, e_n\}$ is the sequence of input embeddings. The attribution score $S(c, R_I)$ for each candidate token $c \in \mathcal{C}_t$ is then computed by Equation 2 by summing relevance over these embeddings. This process selects the token that is most grounded in the instruction part of the prompt.

**Factuality & In-Context Retrieval**  AGD can also be used to improve factual accuracy by defining the ROI over specialized attention heads, leveraging prior work that identifies heads crucial for knowledge processing and retrieval (Jin et al., 2024; Kahardipraja et al., 2025).

- **Closed-Book Factuality:** To reduce hallucinations, the ROI $(R_P)$, is the set of pre-identified parametric knowledge heads. The attribution score $S(c, R_P)$ measures how much the prediction of token $c$ relies on these heads. By maximizing this score, we encourage the model to select tokens based on the factual knowledge encoded within its parameters.

- **Open-Book Retrieval:** To ground the output in provided evidence, we can define the ROI in two ways: (1) as the set of in-context retrieval heads $(R_{IC})$, or (2) as the input embeddings of the context document itself $(R_C = \{e_i \mid x_i \in x_{\text{context}}\})$. Both approaches aim to select tokens that are maximally grounded in the provided evidence.

In Appendix D, we give additional details on the identification process of these specialized heads.

## 3.4 Adaptive Guidance with Entropy-Gating

Applying AGD at every decoding step is computationally expensive due to multiple backward passes of the attribution and can degrade text quality when the model is already confident. To mitigate this, we introduce an adaptive strategy that applies guidance selectively. Motivated by recent work showing that generation trajectories are largely determined by a few high-entropy *critical forks* (Wang et al., 2025), we use the Shannon entropy of the output distribution as a trigger for intervention. Let $H(\boldsymbol{p}_t)$ be the entropy of the probability distribution $\boldsymbol{p}_\theta(y_t \mid x, y_{<t})$. AGD is only applied when the model is uncertain, *i.e.*, when its output entropy exceeds threshold $\tau$. Otherwise, we default to standard greedy decoding. The final selection rule is:

$$y_t = \begin{cases} \arg\max_{c \in \mathcal{V}} \boldsymbol{p}_\theta(c \mid x, y_{<t}) & \text{if } H(\boldsymbol{p}_t) < \tau \\ \arg\max_{c \in \mathcal{C}_t} S(c, R) & \text{if } H(\boldsymbol{p}_t) \geq \tau \end{cases} \tag{5}$$

This entropy-gating mechanism can significantly reduce the computational overhead of AGD while preserving its benefits for instruction adherence, as intervention is focused only on critical decision points where the model is most likely to deviate from the desired behavior.

## 4 Instruction following

To comprehensively evaluate the effectiveness of our decoding approach on instruction following task, we conduct experiments across three instruction-tuned language models: Llama 3.1 (8B) (Grattafiori et al., 2024), Qwen 2.5 (7B) (Yang et al., 2024), and Gemma 3 (4B) (Team et al., 2025). Below, we detail datasets, specify the ROI and the metrics used for evaluation.

### 4.1 Experimental setup

**Datasets and metrics** To assess the instruction-following ability under verifiable constraints, we utilized the *IHEval* rule following dataset (Heo et al., 2025), which is based on the *IFEval* dataset (Zhou et al., 2023a), covering 25 types of constraints. Each example contains a clear separation between the instruction (system prompt) and the task (user prompt). We select IHEval to isolate and control the constraint-following evaluation, avoiding the complexity introduced in the original IFEval, where instructions are embedded within less structured input. For AGD, the ROI is the set of input embeddings corresponding to the system prompt. As evaluation metrics, we report loose **Prompt Level Accuracy (PLA)**, the proportion of outputs satisfying all constraints, along with **Instruction Level Accuracy (ILA)**, as each example can consist of more than one constraint. To measure generation quality, we follow Stolfo et al. (2025) and report a **Quality Score (QS)**, which is a fraction of *yes* answers from an LLM evaluator to yes/no questions about the utility of a response, given that all constraints are satisfied. These questions were first generated by the same evaluator based on a task-only input (excluding constraint). We report details of this procedure in the Appendix B. Finally, we report the combined metric **(PLA * QS)** to balance adherence and quality.

To examine instruction-following in-the-wild, under more complex, multi-turn conversational settings, we leverage the *SysBench* dataset (Qin et al., 2025), a bilingual Chinese-English benchmark containing 500 examples. Each example includes a system prompt with complex constraints and five subsequent user-model turns. The ROI for AGD is the entire system prompt across whole conversation. In line with Qin et al. (2025), we report three metrics: **Constraint Satisfaction Rate**

Table 1: Performance on instruction following benchmarks. Higher is better for all metrics (%). AGD subscripts denote the attribution method (I×G or LRP) and whether it is entropy-gated (e). **PLA**: Prompt-Level Accuracy, **ILA**: Instruction-Level Accuracy, **QS**: Quality Score. **CSR**, **ISR**, and **SSR** are composite metrics for the multi-turn SysBench task.

| Model | Method | IHEval | | | SysBench | | |
|-------|--------|-----------|------|--------|------|------|------|
| | | PLA (ILA) | QS | PLA*QS | CSR | ISR | SSR |
| Llama 3.1 (8B) | Greedy | 66.0 (75.8) | 81.3 | 53.7 | 67.1 | 48.4 | 26.0 |
| | Nucleus | 63.6 (73.3) | 73.9 | 47.0 | 58.0 | 40.6 | 20.2 |
| | CAD | 73.9 (81.3) | 72.6 | 53.7 | 72.2 | **58.8** | 32.3 |
| | AGD$_{I×Ge}$ | 67.1 (76.9) | **82.1** | 55.1 | 67.8 | 50.1 | 27.2 |
| | AGD$_{I×G}$ | 70.8 (79.6) | 81.8 | **57.9** | 65.1 | 46.5 | 24.2 |
| | AGD$_{LRPe}$ | 74.5 (82.6) | 76.4 | 56.9 | **74.3** | 58.2 | **33.9** |
| | AGD$_{LRP}$ | **79.1 (85.0)** | 73.2 | **57.9** | 73.3 | 57.3 | 32.2 |
| Qwen 2.5 (7B) | Greedy | 63.2 (72.7) | 74.1 | 46.8 | 67.6 | 47.9 | 27.1 |
| | Nucleus | 62.8 (72.7) | 75.2 | 47.2 | 64.8 | 44.8 | 24.7 |
| | CAD | 67.3 (76.6) | 67.4 | 45.4 | 65.7 | 49.2 | 25.2 |
| | AGD$_{I×Ge}$ | 62.5 (72.5) | **75.9** | 47.4 | 67.3 | 46.9 | 25.1 |
| | AGD$_{I×G}$ | 65.6 (74.2) | 74.8 | 49.1 | 66.8 | 46.4 | 25.2 |
| | AGD$_{LRPe}$ | **70.4 (78.3)** | 70.6 | **49.7** | 71.1 | 53.0 | 29.9 |
| | AGD$_{LRP}$ | 70.1 (**78.5**) | 67.4 | 47.2 | **73.7** | **56.4** | **32.7** |
| Gemma 3 (4B) | Greedy | 84.7 (89.8) | 82.3 | 69.7 | 69.8 | 52.4 | 33.3 |
| | Nucleus | 83.3 (88.9) | 85.2 | 71.0 | 69.3 | 52.2 | 33.2 |
| | CAD | 81.0 (87.1) | 73.2 | 59.3 | 73.0 | **57.9** | 36.0 |
| | AGD$_{I×Ge}$ | 83.0 (88.7) | **87.3** | **72.5** | 69.0 | 51.6 | 32.2 |
| | AGD$_{I×G}$ | 80.6 (86.9) | 86.6 | 69.8 | 68.5 | 50.4 | 31.8 |
| | AGD$_{LRPe}$ | **86.7 (91.0)** | 81.4 | 70.6 | 73.0 | **57.9** | 36.0 |
| | AGD$_{LRP}$ | 86.0 (90.5) | 78.4 | 67.4 | **73.2** | 57.8 | **36.5** |

**(CSR)**, measuring the average proportion of satisfied constraints; **Instruction Satisfaction Rate (ISR)**, measuring the proportion of individual responses fully satisfying constraints; and **Session Stability Rate (SSR)**, measuring the average number of consecutive turns satisfying all constraints from the conversation's start. Responses are evaluated exclusively by an LLM with respect to the system prompt constraints, thus blending adherence and utility metrics.

**Baselines** We compare AGD against standard decoding methods – **greedy** and **nucleus sampling** ($p = 0.95$) – and the stronger baseline **Context-aware Decoding (CAD)** (Shi et al., 2024), a method that modifies output logits via contrastive decoding between a prompt with and without the instruction, adapted to improve adherence. For CAD we set the control hyperparameter $\alpha = 1$.

**Settings** To form the candidate set $C_t$ (Section 3.2), we apply a top-$k$ constraint and a minimum probability threshold $\pi_{\min}$. This design ensures that $C_t$ remains small and focused, so that attributions are computed only over semantically plausible candidates. To ensure fair comparison and demonstrate the generality of the method, we fix the hyperparameters across all experiments, setting $k = 5$ and $\pi_{\min} = 0.05$. For our entropy-gated variants, we set the activation threshold to $\tau = 1.734$. This value corresponds to the 80th percentile of token-level entropy on IHEval and is motivated by prior work on identifying critical generation steps (Wang et al., 2025) (see Appendix C).

## 4.2 RESULTS

As shown in Table 1, our method significantly improves instruction adherence on both datasets. On IHEval, AGD with LRP attribution (AGD$_{LRP}$) consistently achieves the highest Prompt-Level Accuracy (PLA), boosting it by 13.1 points for Llama 3.1 over greedy decoding. While this strong guidance can lower the Quality Score (QS), the entropy-gated version (AGD$_{LRPe}$) effectively mitigates this trade-off, preserving higher quality while retaining most of the adherence gains. Overall,

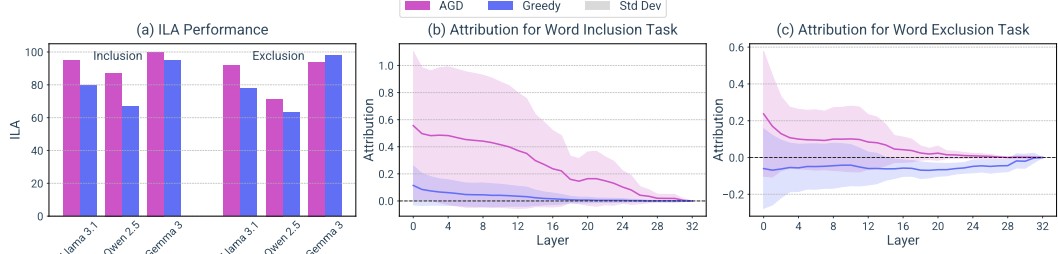

Figure 2: Analysis of attribution signal for word inclusion and exclusion tasks. (a) AGD improves performance on both task types. (b, c) Layer-wise attribution in the residual stream for Llama 3.1 (8B) at decision points where AGD's choice diverges from the greedy path to satisfy a constraint.

entropy-gated variants consistently improve QS compared to their basic counterparts, with a notable trade-off in instruction adherence observed only for Llama 3.1. I×G attribution ($AGD_{I\times G}$) preserves or even enhances quality over greedy decoding, but does not consistently improve adherence. [1]

On the more complex multi-turn SysBench benchmark, AGD's advantages persist, particularly in maintaining long-term adherence. For example, with Llama 3.1, $AGD_{LRPe}$ improves the Session Stability Rate (SSR) by 7.9 points, showing a substantial increase in the model's ability to remember and follow initial instructions over multiple turns. While the CAD baseline is sometimes competitive on the ISR metric, AGD variants consistently show superior performance across others. Overall, LRP proves to be a more effective attribution method than the simpler I×G, providing a more robust mechanism for guiding generation toward instruction adherence.

## 4.3 ANALYSIS & CASE STUDIES

To illustrate how AGD operates, we visualize the attribution scores of candidate tokens for different instruction types in Figure 3. We observe that across various tasks – including word inclusion and exclusion (a, b), length manipulation (c, d), and format adherence (e) – tokens that satisfy the given instruction consistently exhibit higher attribution scores within the relevant parts of the prompt.

**The Role of the Attribution Sign**  Attribution methods often produce both positive and negative scores, which provide distinct and valuable guidance signals. This is particularly evident when comparing two distinct instruction types from IHEval: keyword existence (e.g., `Your response must include the keywords 'forests' and 'riddle'`) and forbidden words (e.g., `Do not mention the words 'Taylor', 'Swift', or 'Together'`). As shown in Figure 2a, AGD successfully improves adherence for both positive (inclusion) and negative (exclusion) constraints (except on Gemma 3 (4B), where forbidden words baseline performance is already near-saturated). On average, a token that satisfies an inclusion rule exhibits a stronger positive attribution signal on instruction inputs throughout the residual stream of the model's layers (Figure 2b), a process exemplified in Figure 3a where the candidate token "intern" receives high positive attribution from the same token in the instruction.

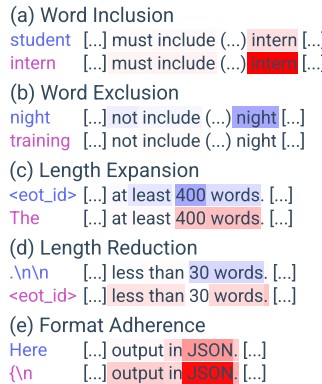

Figure 3: Heatmap visualizations of attribution scores on input token embeddings. For each task, we compare a token that adheres to the instruction (adherent) with one that does not (non-adherent). Spaces are omitted for clarity. Full examples are in Appendix H.

In contrast, for negative constraints, candidates must be suppressed. When a forbidden word appears as a candidate, it exhibits a negative attribution signal on the instruction inputs, a penalty observed on average across all layers (Figure 2c). For instance, as shown

---

[1]Note that QS is measured only on samples where instructions are fully met; methods with lower PLA are thus evaluated on a potentially easier subset of examples, which may inflate their QS.

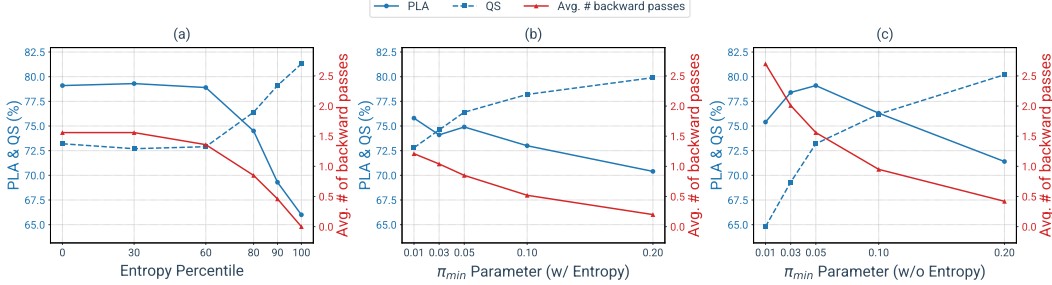

Figure 4: Ablation analysis for Llama 3.1 (8B)

Table 2: Qualitative comparison of model outputs for the riddle example. Tokens where AGD was activated (i.e., entropy exceeded $\tau$) are highlighted in blue. For standard AGD (no entropy), all generated tokens are highlighted. More examples in Appendix C.

| Example | Method | Output | Quality Evaluation |
|---|---|---|---|
| **Instruction:** Your response must include the keywords 'afternoon' and 'distressed'. Do not use any commas in your response.

**Task:** Write a riddle about embroidery that has the answer 'needle'. | Greedy | I'm used to stitch and to sew\nIn the afternoon I'm often in view\nA tool for the crafty and skilled\nHelping to create with a delicate will\nI'm small but my work is grand\nA tiny hero in a crafty hand\n\nWhat am I?
Constraint failed: 'distressed' keyword not used. | [...] **4. Question:** Is the riddle grammatically correct and well-structured? **Reason:** The riddle is grammatically correct and follows a coherent structure, making it easy to read and understand. **Answer:** Yes [...] |
| | AGD$_{LRP}$
(w/o entropy) | In the afternoon I am carefully chosen to weave a delicate design on fabric with threads so fine I pierce through layers one stitch at a time leaving a path of yarn that is both intricate and a work of art often causing the crafter to become distressed when they lose their thread | [...] **Reason:** The response is a single, long sentence that lacks punctuation, making it difficult to read and understand. **Answer:** No [...] |
| | AGD$_{LRPe}$
(w/ entropy) | I pierce the fabric with gentle care\nAfternoon sunbeams highlight my delicate thread there\nA tool for artisans both skilled and distressed\nI guide the yarn through intricate designs I've guessed. | [...] **Reason:** The riddle is grammatically correct and uses a consistent poetic structure, making it well-structured. **Answer:** Yes [...] |

in Figure 3b, the token "night" exhibits negative attribution with respect to itself in the instruction prompt. This negative score penalizes its selection, allowing an alternative, correct token to be chosen instead, even if its own positive attribution score is small. This demonstrates that negative attribution scores offer a strong signal for avoidance, making them crucial for steering models away from undesirable outputs. Consequently, only attribution methods that produce negative signals are applicable in this context.

**Qualitative Impact of Entropy-Gating** As established in our main results, constant intervention during decoding can degrade output quality. Our entropy-gated AGD mitigates this by applying guidance only at decision points where the model is uncertain. This creates a trade-off, quantified in Figure 4a: more frequent intervention boosts instruction adherence (PLA) but can degrade output quality (QS). Our chosen 80th percentile threshold strikes a balance, capturing most of the adherence gains while preserving quality and reducing computational overhead. The detailed analysis of the computational efficiency is presented in the Appendix E.

Table 2 provides a qualitative illustration of this trade-off. In the example, the greedy-decoded riddle is well-formed but fails the instruction by omitting the keyword "distressed". The standard AGD output, which intervenes at each decoding step, includes the keyword but produces a single, ungrammatical run-on sentence. In contrast, the entropy-gated version successfully adheres to all constraints while maintaining grammatical correctness and a coherent structure. This highlights that entropy-based intervention is essential for robust instruction adherence without harming quality.

**Ablations** We ablate the minimum probability threshold $\pi_{min}$ used to form $C_t$ (Section 4.1) to analyze its impact on adherence, quality, and efficiency (Figure 4b,c). A lower $\pi_{min}$ expands the candidate set, which can improve adherence but becomes detrimental at extremely low values. We hypothesize this is because the set becomes polluted with noisy, low-probability tokens that may be selected due to spurious high attribution scores, degrading quality and increasing computational cost. Conversely, a high $\pi_{min}$ improves efficiency but causes adherence to drop sharply as correct tokens

are prematurely filtered out. Our experiments show that $\pi_{\min} = 0.05$ provides a robust balance, enabling high performance without being computationally prohibitive or susceptible to noise.

## 5 FACTUALITY & IN CONTEXT RETRIEVAL

To demonstrate the versatility of AGD beyond instruction following, we evaluate it on knowledge-intensive Question Answering (QA) in two distinct settings. In the **closed-book** setting, the model must answer questions using only its internal, parametric knowledge. Here, the goal is to mitigate hallucinations by steering generation to rely on the components responsible for storing factual information. In the **open-book** setting, the model is provided with a context document containing the answer. The goal is to improve its ability to accurately ground its response in the provided evidence.

### 5.1 EXPERIMENTAL SETUP

For the **closed-book** setting, we guide generation by maximizing attribution towards a pre-identified set of parametric knowledge heads (AGD$_{\text{LRPh}}$). For the **open-book** setting, we explore two guidance strategies: maximizing attribution towards the input embeddings of the provided context (AGD$_{\text{LRPc}}$), or towards a pre-identified set of in-context heads responsible for contextual processing (AGD$_{\text{LRPh}}$). We evaluate on three standard QA benchmarks: MRQA version (Fisch et al., 2019) of TriviaQA (TQA) (Joshi et al., 2017) and Natural Questions (NQ) (Kwiatkowski et al., 2019), as well as Hot-PotQA (HPQA) (Zhilin et al., 2018), all of each can be processed with or without the context. We report recall (Adlakha et al., 2024) as our performance metric, since instruction-tuned models tend to produce verbose outputs.

To assess AGD across different retrieval quality scenarios, we employ two evaluation paradigms. For TQA and NQ, we provide a gold paragraph containing the answer as context, which makes them a standard RAG baselines where the retrieval step is replaced by oracle context provision. This setup isolates AGD's ability to utilize provided evidence from retrieval quality concerns. To assess robustness in realistic RAG scenarios where retrieved contexts contain noise, we use the HPQA *distractor* split, which pairs each question with 2 gold and 8 distractor paragraphs, all shuffled randomly. This simulates imperfect retrieval and tests whether AGD's attribution mechanism provides value beyond oracle retrieval settings. Details of the data preprocessing can be found in the Appendix A.

Our baselines include greedy decoding and nucleus sampling, supplemented by strong, task-specific methods. For the closed-book setting, we add DoLA (Chuang et al., 2024), which is designed to reduce hallucinations by contrasting logits from different model layers. For the open-book setting, we use again Context-aware Decoding (CAD) (Shi et al., 2024), as its contrastive mechanism is specifically designed to ground generation in a provided context.

### 5.2 RESULTS

The results are presented in Table 3. In the closed-book setting, guiding generation towards parametric knowledge heads (AGD$_{\text{LRPh}}$) improves factual recall. For Llama 3.1 (8B), it outperforms standard decoding and the DoLA baseline on TQA and HPQA. This trend holds for the Qwen 2.5 (7B) model, though improvements are less pronounced for the smaller Gemma 3 (4B). In the open-book setting, AGD yields more consistent and significant gains across all models. Guiding generation towards either the provided context embeddings (AGD$_{\text{LRPc}}$) or in-context heads (AGD$_{\text{LRPh}}$) consistently outperforms baselines across all datasets and models, with the context-embedding strategy generally proving slightly more effective. Notably, even for Gemma 3 (4B), where closed-book improvements were limited, AGD provides a clear boost in performance, demonstrating its effectiveness at grounding generation in provided evidence.

Table 3: Recall score (%) of Llama 3.1 (8B) in both closed-book (top) and open-book (bottom) settings. Higher scores are better. Full results are in Appendix F.

| Method | TQA | NQ | HPQA |
|---|---|---|---|
| Greedy | 81.4 | 63.6 | 34.6 |
| Nucleus | 79.0 | 59.9 | 31.9 |
| DoLA | 81.2 | **63.8** | 34.3 |
| AGD$_{\text{LRPh}}$ | **82.4** | 63.0 | **39.6** |
| Greedy | 89.4 | 83.5 | 81.3 |
| Nucleus | 89.7 | 83.3 | 80.7 |
| CAD | 87.9 | 84.6 | 83.7 |
| AGD$_{\text{LRPh}}$ | 91.0 | 87.0 | **87.9** |
| AGD$_{\text{LRPc}}$ | **91.4** | **87.9** | **87.9** |

The HPQA results demonstrate AGD's effectiveness even with 80% distractor content. For Llama 3.1 (8B), both variants of AGD achieve an improvement of 6.6 points compared to greedy decoding. It suggests that the attribution mechanism helps models ground generation in relevant portions rather than being misled by noise. To verify that these gains come from the attribution mechanism rather than prompt engineering (Sclar et al., 2024), we tested an alternative prompt formulation (see Appendix G). AGD maintains consistent advantages across both prompt variants, confirming the robustness of our approach. Overall, these results show that AGD is a potent method for enhancing the factual accuracy and contextual grounding of LLMs, with particularly strong performance in open-book retrieval scenarios, including realistic settings with imperfect retrieval.

## 6 CONCLUSION

In this work, we introduced Attribution-Guided Decoding (AGD), a fine-tuning-free decoding strategy that enhances LLM reliability by selecting tokens that maximally attribute to a specified Region of Interest (ROI), such as a user instruction or a knowledge-storing component. Our experiments demonstrate that this approach significantly improves both instruction adherence and factual accuracy in closed-book and open-book settings, while an entropy-gated variant preserves output quality and reduces computational cost by applying guidance selectively.

AGD's primary limitation is inherent to its design as a selection mechanism: it cannot generate a desired token if it is not proposed by the model. Other challenges include the computational cost of multiple backward passes and the need to define a relevant ROI for each task. Future work could focus on developing more efficient attribution proxies to mitigate these costs. Moreover, the ROI concept could be extended from input spans or attention heads to more monosemantic structures, such as specific, functionally-identified circuits within the model, enabling more granular control. The code is available at `https://github.com/piotr-komorowski/attribution-guided-decoding`.

## ACKNOWLEDGMENTS

We would like to express our gratitude to Melina Zeeb for her valuable assistance in preparing the main figure of this paper, and to Patrick Kahardipraja for his constructive feedback on this paper. This work was in part supported by the Federal Ministry of Research, Technology and Space (BMFTR) as grant xJuRAG (16IS25015B); the German Research Foundation (DFG) as research unit DeSBi [KI-FOR 5363] (459422098); and the European Union's Horizon Europe research and innovation programme (EU Horizon Europe) as grant TEMA (101093003).

## REPRODUCIBILITY STATEMENT

The source code for Attribution-Guided Decoding (AGD) and all experimental scripts will be made publicly available upon publication. We provide detailed descriptions of our experimental setup throughout the paper, including the specific models used (Section 4), datasets and the fixed hyperparameters for both AGD and all baselines (Sections 4.1 and Appendix A). All experiments involving randomness, such as nucleus sampling, were conducted with a fixed random seed to ensure consistent outcomes. Further implementation details, including the exact prompt templates used for data preprocessing steps (Appendix A), the quality scoring protocol (Appendix B), and the methodology for extracting specialized attention heads (Appendix D), are documented in the Appendix.

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

## A  GENERAL IMPLEMENTATION

Method and experiments were implemented using PyTorch (Paszke et al., 2019). LLM-based evaluations for Quality Score (QS) for IHEval and SysBench metrics were performed using gpt-4o-2024-08-06.

**Datasets**  For the IHEval dataset, we used the `rule-following/single-turn/aligned` split from its official repository (https://github.com/ytyz1307zzh/IHEval). The SysBench dataset was sourced from its original repository (https://github.com/PKU-Baichuan-MLSystemLab/SysBench). For TriviaQA (TQA) and Natural Questions (NQ), we used the development splits from the MRQA 2019 Shared Task repository (https://github.com/mrqa/MRQA-Shared-Task-2019). Lastly, for HotPotQA (HPQA), we used the validation split from Hugging Face (https://huggingface.co/datasets/hotpotqa).

Factuality datasets underwent a preprocessing procedure. Specifically for NQ, we applied the steps proposed by Yu et al. (2024). Additionally, we filtered out duplicate entries from each dataset, performing this process independently for the closed-book (CB) and open-book (OB) settings. This filtering accounts for instances in the OB setting where the same question may be paired with different evidence passages, but also when context was empty. This process resulted in final evaluation sets of 7,785 samples for TQA (CB and OB); 4,987 for NQ (CB) and 5,450 (OB); and 5,918 for HPQA (CB and OB).

For all experiments, we used the system prompt: `You are a helpful assistant.` For closed-book settings, the user prompt consisted solely of the question. For open-book settings, we used the following prompt structure: `{{ context }} \n\nBased on this text, answer this question:\nQ: {{ question }}\nA:`.

**Baselines**  Proposed by Shi et al. (2024), CAD is a contrastive decoding method designed to improve the faithfulness of generation to a given context. It modifies the output logits at each step by amplifying the difference between the distribution conditioned on the full input and a distribution conditioned on a partial, "context-free" input. The modified logit is computed as:

$$\text{logit}'(y_t) = (1 + \alpha) \cdot \text{logit}(y_t \mid x_{\text{full}}) - \alpha \cdot \text{logit}(y_t \mid x_{\text{context-free}}) \qquad (6)$$

We use CAD as a baseline for both instruction following (where $x_{\text{full}}$ includes the instruction and $x_{\text{context-free}}$ omits it) and open-book QA (where $x_{\text{context-free}}$ omits the provided document). Following the original work, as one of the plausible choices, we set the control hyperparameter $\alpha = 1.0$.

Proposed by Chuang et al. (2024), DoLA is a method designed to reduce hallucinations in closed-book settings. It is based on the finding that factual knowledge in transformers is often localized in specific layers. The method works by modifying the output logits at each decoding step – it contrasts the logits from the final layer with logits projected from one of earlier layers, exploiting the hierarchical encoding of factual knowledge within LLMs. While the original implementation suggests contrasting with *higher* layers for QA tasks, we empirically found that contrasting the final layer with *lower* layers consistently yielded better recall scores across all models and datasets in our setup; we therefore report this.

## B  QUALITY EVALUATION

To evaluate the generation quality for the IHEval dataset, we follow the procedure introduced by Stolfo et al. (2025). First, using only the task portion of the prompt (*i.e.,* without the instruction), we prompt an LLM evaluator to generate up to five simple yes/no questions that break down the core requirements of the task. The examples of these questions can be found in Table 2 and 6. The prompt used for this step is shown in Table 4.

Second, the model's response to the full prompt (task and instruction) along with task itself is evaluated against these generated questions. The evaluator is prompted to answer *Yes*, *No* or *Not Applicable* for each question, providing a brief justification. The prompt for this evaluation step is shown in Table 5. he final Quality Score (QS) is defined as fraction of *Yes* responses out of *Yes* and *No*, calculated only for responses that successfully satisfied all instructions.

## C  ENTROPY-GATING DETAILS

The entropy threshold ($\tau = 1.734$) for our adaptive AGD variant was chosen based on the distribution of token-level entropy observed on the IHEval dataset, as shown in Figure 5. This value corresponds to the 80th percentile.

```
The following is a prompt that is used to evaluate the generations from a large
language model. We do not know how to evaluate the quality of model answers for this
prompt. Can you come up with 5 or less questions that can break down the quality to
simpler evaluation tasks that we can then ask about the model answer? Each question
should have a simple yes, no answer.
Prompt: {{ prompt without instruction }}
List all sub questions in the following format:
Output:
1: Question: <question>
2: Question: <question>
...
N: Question: <question>
```

Table 4: Prompt for Quality Evaluation Question Generation.

```
We need to evaluate the quality of generations from a large language model. You will
be given an input prompt, the response from a language model and a set of questions
assessing the quality of the response. You need to review the response against the
input prompt and provide an answer to each question as either 'Yes', 'No' or 'Not
Applicable' if the question does not apply to the case along with a reason for your
answer.
Prompt: {{ prompt without instruction }}
Response: {{ response }}
Questions: {{ up to 5 evaluation questions }}
List your answers in the following format:
Output:
1. Question: <question>. Reason: <reason>: Answer: <answer>
2. Question: <question>. Reason: <reason>: Answer: <answer>
...
N. Question: <question>. Reason: <reason>: Answer: <answer>
```

Table 5: Prompt for Quality Score Evaluation

Another example of how entropy-gated version of AGD handles adherence–quality trade-off is presented in Table 6. Greedy decoding fails to include enough placeholders. Standard AGD satisfies the constraints but overuses placeholders, resulting in an unnatural and poorly formatted email. The entropy-gated output again finds a superior balance, meeting the placeholder requirement while preserving the fluency and structure of a natural email.

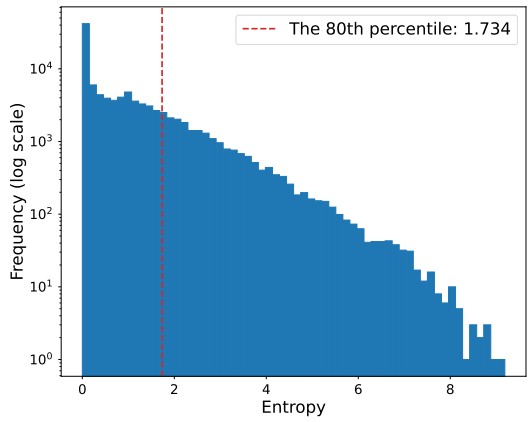

Figure 5: Distribution of token entropy on IHEval dataset for Llama 3.1 (8B)

Table 6: Qualitative comparison of model outputs for the email example. Tokens where AGD was activated (i.e., entropy exceeded $\tau$) are highlighted in blue. For standard AGD (no entropy), all generated tokens are highlighted.

| Example | Method | Output | Quality Evaluation |
|---|---|---|---|
| **Instruction:** Your response must contain at least 10 placeholders represented by square brackets, such as [name].

**Task:** Rewrite the following sentence into an email: The boots are warm but a little uncomfortable. | Greedy | Here is the rewritten sentence as an email:\n\nSubject: Feedback on [brand] Boots\n\nDear [name],\n\nI wanted to follow up with you regarding the [color] [brand] boots I received. I have been wearing them for [length of time] and I have to say that they are [adjective 1] but a little [adjective 2]. I was hoping to get your thoughts on this and see if there's anything that can be done to improve the fit.\n\nBest,\n[Your Name]
Constraint failed: not enough placeholders. | [...] **2. Question:** Is the rewritten sentence formatted appropriately for an email? **Reason:** The response includes a subject line, a greeting, and a closing, which are typical components of an email format. **Answer:** Yes [...] |
| | AGD$_{LRP}$ (w/o entropy) | Here is the rewritten sentence as an email:\n\nSubject: [name] - Boot Review\n\nDear [name],\n\nI wanted to share with you my thoughts on the [brand] boots that [name] bought [time] ago. Overall, they seem to be [adjective] and [adjective] so far, but I have found them to be [adjective] at times. Specifically, they can be [adjective] and [adjective], making it [verb] for [name] to [action] for [time]. \n\nBest regards,\n[Your name] | [...] **Reason:** The response includes a subject line and a greeting, which are typical components of an email format. However, the placeholders make it incomplete and not fully formatted. **Answer:** No [...] |
| | AGD$_{LRPe}$ (w/ entropy) | Here is the rewritten sentence as an email:\n\nSubject: [name]'s Boot Review\n\nDear [name],\n\nI wanted to follow up with you regarding the [brand] boots that [name] purchased for [occasion]. The boots are [adjective1] [adjective2], but [name] them to be a bit [adjective3] at times.\n\nIf you have any concerns or would like to discuss further, please don't hesitate to reach out.\n\nBest regards,\n[Your Name] | [...] **Reason:** The response includes a subject line and a greeting, which are typical components of an email format. However, the placeholders like [name], [brand], [occasion], and [adjective] need to be replaced with actual content for the email to be complete. **Answer:** Yes [...] |

# D EXTRACTION OF FACTUALITY & IN-CONTEXT HEADS

Following the methodology of Kahardipraja et al. (2025), we aim to identify sets of in-context heads $\mathcal{H}_{\mathrm{ctx}}$, that retrieve contextual information, and parametric heads $\mathcal{H}_{\mathrm{param}}$, that store the factual memory of the model. In-context heads are defined as those contributing mainly in open-book settings by retrieving contextual information, whereas factual heads dominate in closed-book conditions by relying on internal parametric knowledge. Each head type is maximally influential in its respective setting while having minimal effect in the other. To extract the heads, we analyze counterfactual contexts from the NQ-Swap dataset (Longpre et al., 2021). First, open-book questions with counterfactual contexts are presented to the model, producing predictions $c_{\mathrm{cf}}$ that are guaranteed to be absent from the model's internal knowledge due to the counterfactual nature of the context. Next, closed-book questions, where contextual information is minimized, are used to isolate the model's parametric components, yielding parametric predictions $c_{\mathrm{gold}}$.

Let $\mathbf{X} = (\mathbf{x}_1, \ldots, \mathbf{x}_S) \in \mathbb{R}^{d \times S}$ denote the matrix of hidden token representations for a sequence of length $S$ with dimension $d$, and suppose our model employs $H$ parallel heads, each of dimension $d_h = d/H$. Then, the computation of the multi-head attention layer can be reformulated into $H$ complementary operations, where each head $h$ produces an intermediate attention output $\mathbf{z}_i^h \in \mathbb{R}^{d_h}$:

$$\mathbf{z}_i^h = \sum_{j=1}^{S} \mathbf{A}_{i,j}^h (\mathbf{W}_V^h \mathbf{x}_j) \tag{7}$$

We use AttnLRP to quantify head contributions by summing relevance scores of each head's latent output $\mathbf{z}^h$ across tokens and dimensions:

$$r_h(c) = \sum_{i=1}^{S} \sum_{k=1}^{d_h} \mathcal{A}_\theta(c \mid x, y_{<t}; \mathbf{z}_i^h)_k. \tag{8}$$

To contrast behaviors across settings, we compute a difference score $\mathcal{D}$ representing each head's average relevance in open- versus closed-book conditions:

$$\mathcal{D} = \{ \mathbb{E}_{X_{\mathrm{OB}}}[r_h(c_{\mathrm{cf}})] - \mathbb{E}_{X_{\mathrm{CB}}}[r_h(c_{\mathrm{gold}})] \ : \ h = 1, \ldots, N_h \}. \tag{9}$$

We then select the top $N$ heads with the highest and lowest $\mathcal{D}$ values to form $\mathcal{H}_{\mathrm{ctx}}$ and $\mathcal{H}_{\mathrm{param}}$:

$$\mathcal{H}_{\mathrm{ctx}} = \{\mathrm{argsort}_{\mathrm{desc}}(\mathcal{D})\}_{n=1}^{N}, \qquad \mathcal{H}_{\mathrm{param}} = \{\mathrm{argsort}_{\mathrm{asc}}(\mathcal{D})\}_{n=1}^{N}. \tag{10}$$

The $N$ is equal to 100 for Llama 3.1 (8B), 75 for Qwen 2.5 (7B), and 25 for Gemma 3 (4B).

To validate our choice of head count, we evaluated AGD performance with varying numbers of selected heads. The results are presented in Table 7.

Table 7: Ablation study of head count on TriviaQA using Llama 3.1 (8B). Recall scores (%) for closed-book (factuality) and open-book (in-context) settings with different numbers of selected heads.

| # Heads | Factuality | In-Context |
|---------|------------|------------|
| 10 | 81.8 | 90.3 |
| 50 | 81.4 | 91.4 |
| 100 | 82.4 | 91.0 |
| 200 | 82.4 | 90.9 |

The performance is relatively stable across head counts from 50-200. Very small head sets (N=10) underperform – likely we consider too little number of heads and coverage insufficiently the relevant information pathways. We observe no significant degradation with larger head sets, suggesting that the sorting effectively prioritizes relevant heads. We suspected that too big of a number could introduce noise, but it doesn't seem to be the case. Overall, our choice of 100 heads ($\sim$10% of total heads) represents a reasonable middle ground.

## E  COMPUTATIONAL EFFICIENCY

Table 8 presents comprehensive efficiency metrics. The entropy-gated version (LRPe) provides a quite modest increase in compute ranging from 1.32 to 1.6$\times$ compared to greedy decoding, while non-gated requires 1.59-2.20$\times$ more time. This is comparable to the overhead of contrastive methods like CAD (1.32-1.97$\times$), however our method requires more memory. This overhead comes, however, with performance gains, as shown by our results throughout the paper.

## F  FULL FACTUALITY & IN-CONTEXT RETRIEVAL RESULTS

Table 9 presents the complete set of results for the factuality and in-context retrieval experiments across all models, datasets, and methods.

## G  PROMPT SENSITIVITY ANALYSIS

To ensure that AGD's improvements on noisy retrieval are robust to prompt formulation rather than artifacts of specific phrasing (Sclar et al., 2024), we evaluated two distinct prompt structures on the HotPotQA *distractor* split.

We tested two formulations that differ in both structure and emphasis:

- **Standard prompt** (context-first): `{{ context }} \n\nBased on this text, answer this question:\nQ: {{ question }}\nA:`
- **Reordered prompt** (question-first): `Question:  {{ question }} \n\n Based on the text below, provide an answer:\n{{ context }}\nAnswer:`

Both prompts use identical contexts (2 gold paragraphs + 8 distractors, randomly shuffled), but differ in the ordering of question and context as well as the phrasing. The results are presented in Table 10.

AGD maintains consistent advantages over both greedy decoding and CAD across both prompt formulations. The relative ordering of methods remains stable across prompt formulations: AGD variants consistently outperform both greedy and CAD baselines, regardless of whether the context or question appears first, demonstrating robustness to prompt engineering. These results show that AGD's attribution mechanism provides value beyond oracle retrieval settings. It actively helps the

Table 8: Computational efficiency on open-book TriviaQA. AGD subscripts denote whether entropy-gating is used (e). **Time**: average milliseconds per token; **Cost×**: slowdown relative to Greedy; **Memory**: peak GPU memory in MB; **Fwd/Bwd**: average number of forward/backward passes per token. Metrics averaged over 100 samples with max sequence length of 256 tokens on NVIDIA A100 80GB (PyTorch 2.6.0, bf16 precision).

| Model | Method | Time (ms) | Cost× | Memory (MB) | Fwd | Bwd |
|---|---|---|---|---|---|---|
| Llama 3.1 (8B) | Greedy | 94.0 | 1.00× | 15520 | 1 | 0 |
| | CAD | 124.9 | 1.32× | 16215 | 2 | 0 |
| | $AGD_{LRPe}$ | 150.7 | 1.60× | 23238 | 1 | 0.58 |
| | $AGD_{LRP}$ | 207.2 | 2.20× | 23842 | 1 | 1.06 |
| Qwen 2.5 (7B) | Greedy | 85.5 | 1.00× | 14786 | 1 | 0 |
| | CAD | 120.0 | 1.40× | 15566 | 2 | 0 |
| | $AGD_{LRPe}$ | 112.7 | 1.32× | 23628 | 1 | 0.23 |
| | $AGD_{LRP}$ | 161.6 | 1.89× | 23991 | 1 | 0.55 |
| Gemma 3 (4B) | Greedy | 68.6 | 1.00× | 7578 | 1 | 0 |
| | CAD | 134.8 | 1.97× | 9569 | 2 | 0 |
| | $AGD_{LRPe}$ | 92.1 | 1.34× | 14648 | 1 | 0.09 |
| | $AGD_{LRP}$ | 109.0 | 1.59× | 14739 | 1 | 0.34 |

model navigate noisy retrieval results by selecting tokens with stronger grounding in relevant portions of the provided context. These results further validate that the benefit of AGD comes from the decoding mechanism itself.

## H ATTRIBUTION VISUALIZATION DETAILS

This section provides the detailed layer-wise attribution heatmaps that were summarized in Figure 3 of the main paper. For each candidate token, we visualize the relevance scores back-propagated to the input embeddings (Layer 0) and the residual stream of each subsequent transformer layer. To enhance visual clarity, relevance scores at each layer are normalized by the maximum absolute value at that layer. In Figure 3 we omitted the first and the last token from the normalization process to prevent them from dominating the color scale as provided responses can get lengthy and thus token contributions poorly visible. Here we are presenting versions including these tokens. The full input prompt and generation prefix for each example are provided in the figure captions (Figures 6, 7, 8, 9 and 10).

Table 9: Performance of Llama 3.1 (8B), Qwen 2.5 (7B), and Gemma 3 (4B) on TriviaQA, NQ, and HotPotQA datasets in both closed-book and open-book settings, measured in recall (%). Higher scores are better.

| Model | Setting | Method | TriviaQA | NQ | HotPotQA |
|---|---|---|---|---|---|
| Llama 3.1 (8B) | Closed-book | Greedy | 81.4 | 63.6 | 34.6 |
| | | Nucleus | 79.0±0.3 | 59.9±0.3 | 31.9±0.2 |
| | | DoLA | 81.2 | **63.8** | 34.3 |
| | | $AGD_{I \times Gh}$ | 81.2 | 63.0 | 37.5 |
| | | $AGD_{LRPh}$ | **82.4** | 63.0 | **39.6** |
| | Open-book | Greedy | 89.4 | 83.5 | 81.3 |
| | | Nucleus | 89.7±0.3 | 83.3±0.3 | 80.7±0.2 |
| | | CAD | 87.9 | 84.6 | 83.4 |
| | | $AGD_{I \times Gh}$ | 91.2 | 85.7 | 81.6 |
| | | $AGD_{I \times Gc}$ | 89.7 | 83.5 | 83.9 |
| | | $AGD_{LRPh}$ | 91.0 | 87.0 | **87.9** |
| | | $AGD_{LRPc}$ | **91.4** | **87.9** | **87.9** |
| Qwen 2.5 (7B) | Closed-book | Greedy | 69.3 | **47.8** | 33.8 |
| | | Nucleus | 68.8±0.3 | 45.4±0.5 | 32.5±0.3 |
| | | DoLA | 67.9 | 44.3 | 32.4 |
| | | $AGD_{I \times Gh}$ | 69.1 | 47.1 | 33.6 |
| | | $AGD_{LRPh}$ | **70.3** | 46.9 | **34.3** |
| | Open-book | Greedy | 91.1 | 89.0 | 84.5 |
| | | Nucleus | 91.2±0.1 | 88.7±0.2 | 85.0±0.3 |
| | | CAD | 88.6 | 90.0 | 85.0 |
| | | $AGD_{I \times Gh}$ | 91.7 | 90.0 | 85.5 |
| | | $AGD_{I \times Gc}$ | 91.1 | 88.6 | 85.1 |
| | | $AGD_{LRPh}$ | 91.0 | 89.7 | 86.4 |
| | | $AGD_{LRPc}$ | **92.3** | **90.6** | **89.5** |
| Gemma 3 (4B) | Closed-book | Greedy | **61.9** | 41.6 | 27.7 |
| | | Nucleus | 61.3±0.1 | 41.2±0.2 | 27.7±0.2 |
| | | DoLA | 60.5 | 40.8 | 27.3 |
| | | $AGD_{I \times Gh}$ | 61.0 | **42.1** | **28.0** |
| | | $AGD_{LRPh}$ | 61.5 | 41.9 | **28.0** |
| | Open-book | Greedy | 83.2 | 82.0 | 70.1 |
| | | Nucleus | 83.0±0.1 | 82.0±0.1 | 69.8±0.2 |
| | | CAD | 82.0 | 75.7 | 64.1 |
| | | $AGD_{I \times Gh}$ | 83.0 | 82.5 | 70.4 |
| | | $AGD_{I \times Gc}$ | 83.0 | 82.3 | 70.1 |
| | | $AGD_{LRPh}$ | 83.4 | **83.0** | 70.6 |
| | | $AGD_{LRPc}$ | **83.9** | **83.0** | **70.9** |

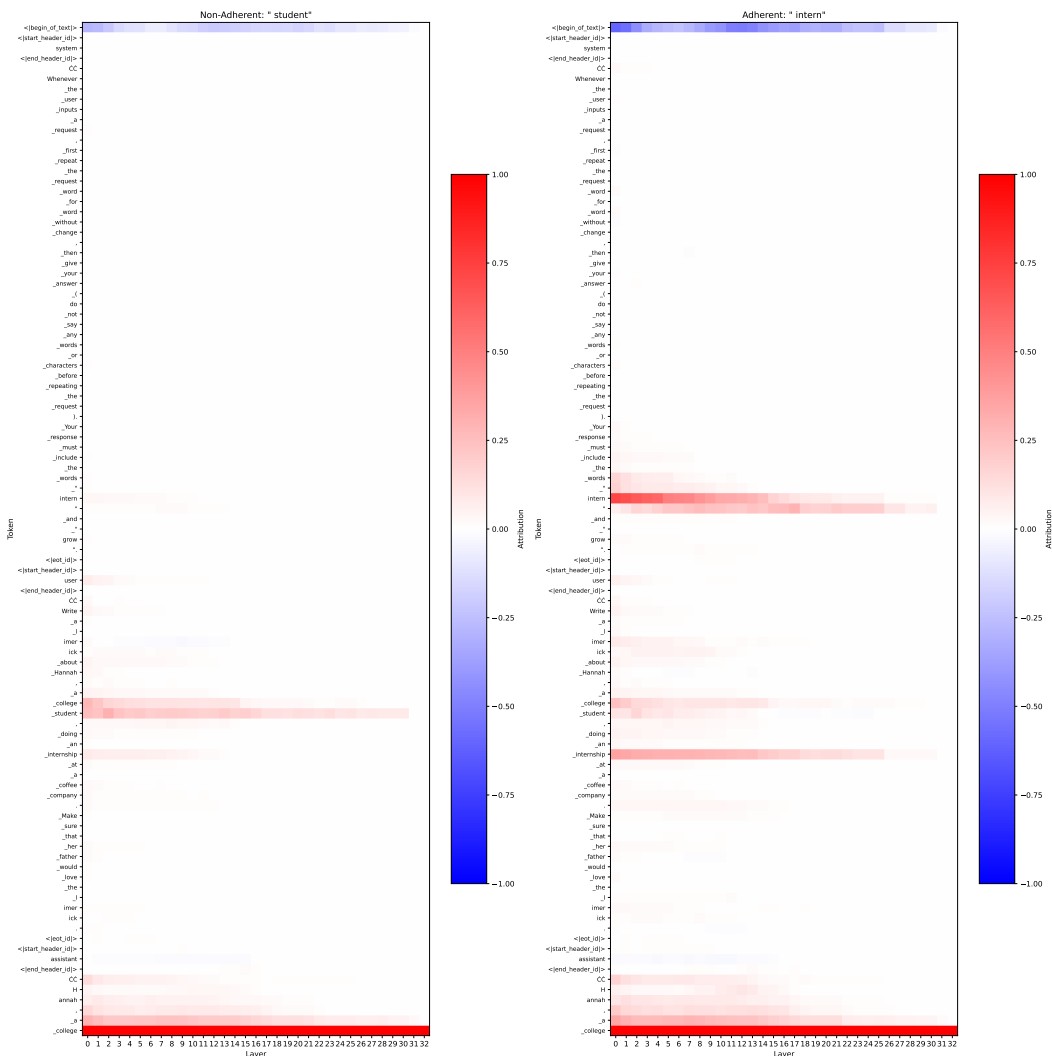

Figure 6: Heatmap example (Word Inclusion). Sequence prefix:
`<|begin_of_text|><|start_header_id|>system<|end_header_id|>\n\n`
`Whenever the user inputs a request, first repeat the`
`request word for word without change, then give your answer`
`(do not say any words or characters before repeating the`
`request).  Your response must include the words "intern" and`
`"grow".<|eot_id|><|start_header_id|>user<|end_header_id|>\n\nWrite`
`a limerick about Hannah, a college student, doing an internship`
`at a coffee company.  Make sure that her father would love the`
`limerick.<|eot_id|><|start_header_id|>assistant<|end_header_id|>`
`\n\nHannah, a college`

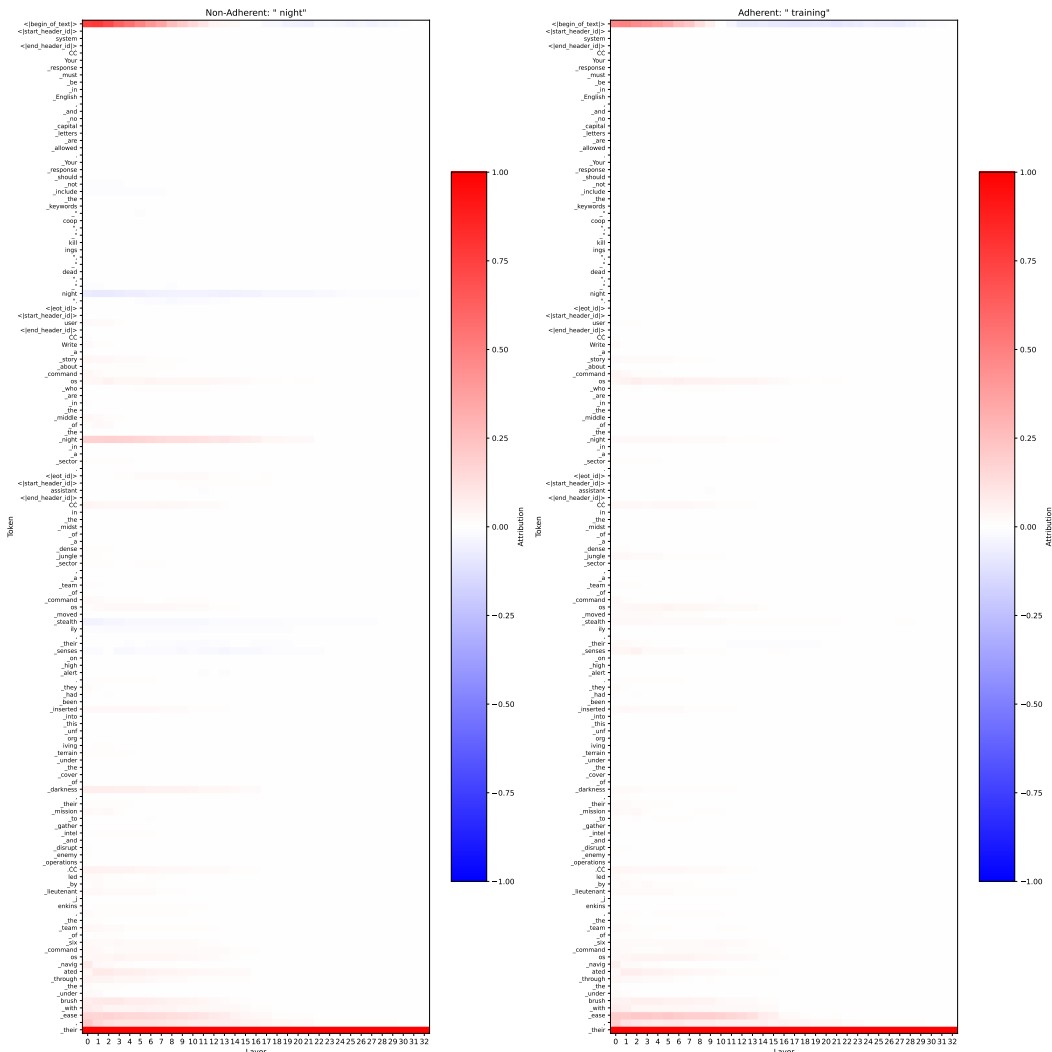

Figure 7: Heatmap example (Word Exclusion). Sequence prefix: `<|begin_of_text|><|start_header_id|>system<|end_header_id|>\n\nYour response must be in English, and no capital letters are allowed. Your response should not include the keywords "coop", "killings", "dead", "night".<|eot_id|><|start_header_id|>user<|end_header_id|>\n\nWrite a story about commandos who are in the middle of the night in a sector.<|eot_id|><|start_header_id|>assistant<|end_header_id|>\n\nin the midst of a dense jungle sector, a team of commandos moved stealthily, their senses on high alert. they had been inserted into this unforgiving terrain under the cover of darkness, their mission to gather intel and disrupt enemy operations.\n\nled by lieutenant jenkins, the team of six commandos navigated through the underbrush with ease, their`

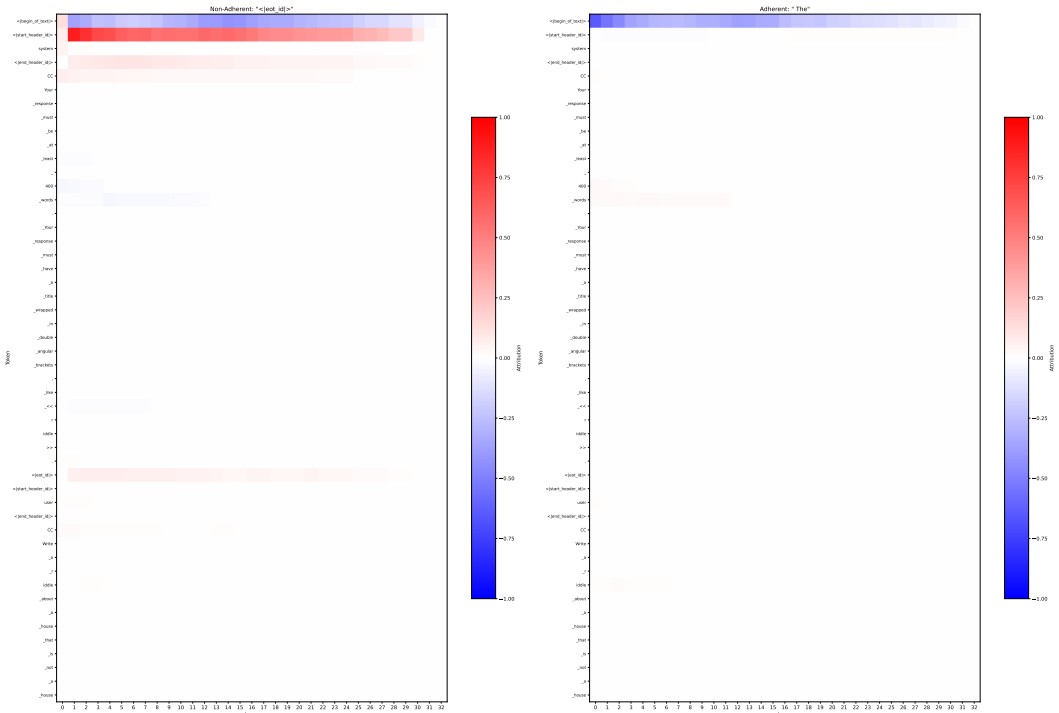

Figure 8: Heatmap example (Length Expansion). Because of the length of the response we limit figure to the 50 initial tokens. Sequence prefix: `<|begin_of_text|><|start_header_id|>system<|end_header_id|>\n\nYour response must be at least 400 words. Your response must have a title wrapped in double angular brackets, like <<riddle>>.<|eot_id|><|start_header_id|>user<|end_header_id|>\n\n Write a riddle about a house that is not a house. Also include the answer.<|eot_id|><|start_header_id|>assistant<|end_header_id|> \n\n<<The House That's Not a House>>\n\nIn the heart of a city, I'm a familiar sight\nA dwelling for none, yet sheltering all day and night\nMy walls are not walls, nor my roof a roof above\nYet I'm where people reside, without a single dispute\n\nMy foundation is solid, though I'm not rooted in earth\nI'm not anchored by beams, nor held together by mirth\nNo mortar nor mortarboard binds me, nor holds me upright\nAnd though I'm not made of sticks, I'm not a fragile delight\n\nI have rooms within me, though none are enclosed\nNo doors swing shut behind you, nor windows to dispense\nNo fireplace crackles warmth within me, though I'm cozy as can be\nNo furniture's arranged within my spaces, though people sit and rest with glee\n\nI shelter from sunbeams bright, though my own light's quite dim\nI shield from raindrops falling, yet my own surface's quite thin\nMy inhabitants don't pay rent, nor utilities to boot\nThough they come and they go freely, without any need to compute\n\nSo, I may seem paradoxical, a riddle indeed\nA house that's not a house, though people reside with ease\nI'm not made of bricks nor blocks, nor crafted with skilled hands\nYet I provide for many needs, though my own needs are quite bland\n\nWhat am I?\n\n Answer:_ A _bus stop_. \n\nA bus stop is indeed a house that's not a house. While it's a dwelling for none in the sense that no person permanently resides within it, a bus stop shelters people waiting for buses all day and night, providing protection from sunbeams and raindrops alike.`

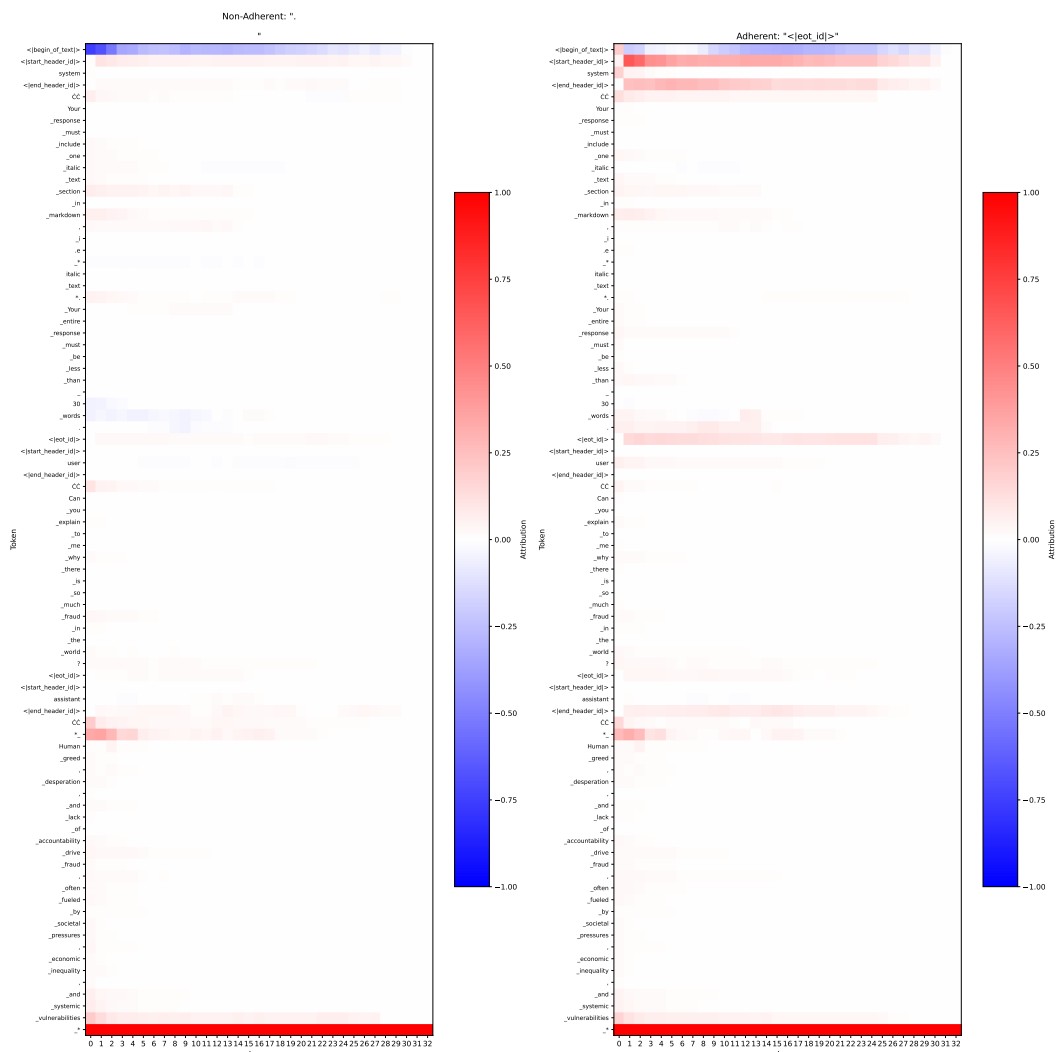

Figure 9: Heatmap example (Length Reduction). Sequence prefix: <|begin_of_text|><|start_header_id|>system<|end_header_id|>\n\nYour response must include one italic text section in markdown, i.e *italic text*. Your entire response must be less than 30 words.<|eot_id|><|start_header_id|>user<|end_header_id|>\n\nCan you explain to me why there is so much fraud in the world?<|eot_id|><|start_header_id|>assistant<|end_header_id|>\n\n*_Human greed, desperation, and lack of accountability drive fraud, often fueled by societal pressures, economic inequality, and systemic vulnerabilities_*

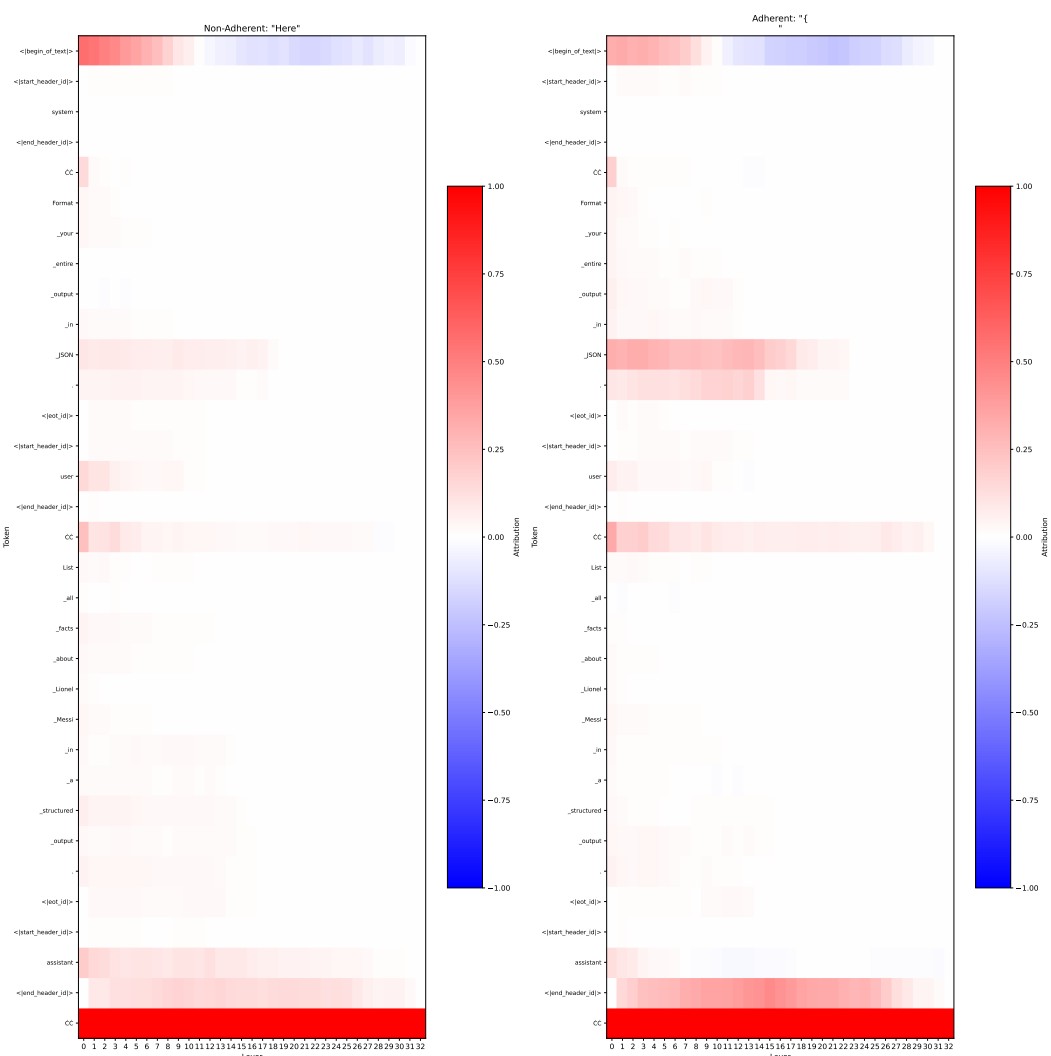

Figure 10: Heatmap example (Format Adherence). Sequence prefix: <|begin_of_text|><|start_header_id|>system<|end_header_id|>\n\n Format your entire output in JSON.<|eot_id|><|start_header_id|> user<|end_header_id|>\n\nList all facts about Lionel Messi in a structured output.<|eot_id|><|start_header_id|>assistant <|end_header_id|>\n\n

Table 10: Recall (%) on HotPotQA *distractor* split with different prompt formulations. AGD maintains consistent advantages across prompt variants despite 80% noise in retrieved contexts.

| Model | Method | Standard Prompt | Reordered Prompt |
|---|---|---|---|
| Llama 3.1 (8B) | Greedy | 81.3 | 82.2 |
| | CAD | 83.7 | 84.1 |
| | AGD$_{LRPh}$ | **87.9** | **87.8** |
| | AGD$_{LRPc}$ | **87.9** | 87.3 |
| Qwen 2.5 (7B) | Greedy | 84.5 | 83.5 |
| | CAD | 85.0 | 81.9 |
| | AGD$_{LRPh}$ | 86.4 | 85.1 |
| | AGD$_{LRPc}$ | **89.5** | **86.0** |
| Gemma 3 (4B) | Greedy | 70.1 | 62.7 |
| | CAD | 64.1 | 53.3 |
| | AGD$_{LRPh}$ | 70.6 | **63.2** |
| | AGD$_{LRPc}$ | **70.9** | 62.9 |

