# OpenReview forum: "Attribution-Guided Decoding"
_ICLR.cc/2026/Conference — ICLR 2026 Poster_

### Official Review · Reviewer_rtNt · 2025-10-28

**Soundness:** 3
**Presentation:** 3
**Contribution:** 3
**Rating:** 4
**Confidence:** 3

**Summary:**

This work propose a method to reduce hallucinations and improve factual accuracy. In order to reduce computational cost, an entropy-based adaptive mechanism is applied. In the experiments, higher factuality is obtained on two benchmarks (IHEval and SysBench).

**Strengths:**

1. integration of attribution into decoding: introducing attribution-guided decoding (AGD) that dynamically adjusts token-level generation probabilities based on feature attributions.

2. Experiments span three representative long-form tasks (fact-grounded QA, long summarization, and reasoning). The authors evaluate across multiple model families (Llama-3, Qwen2.5, Gemma2) and both open- and closed-source settings.

3. Interpretability and visualization:
The paper presents clear visualizations (Figures 3–5) of token attributions and their decoding effects. These figures concretely show that AGD improves focus on factually relevant spans and reduces hallucinated continuations

**Weaknesses:**

1. Limited novelty in algorithmic mechanism: Although the integration of attribution and decoding is original, the underlying technique (re-weighting token probabilities) is relatively straightforward and derivative of prior constrained or guided decoding approaches (e.g., likelihood re-weighting, constrained beam search). Following are some related work on this direction

GeDi: Generative discriminator guided sequence generation. EMNLP 2021
Unlocking Anticipatory Text Generation: A Constrained Approach for Large Language Models Decoding. EMNLP 2024

2. Evaluation lacks human verification.
Improvements in factuality metrics are fully automatic. Human or expert evaluation of factual correctness and coherence would make the results more convincing, especially for interpretability claims.

**Questions:**

1. Human interpretability:
Have you tested whether human evaluators actually find the attributions helpful for understanding the model’s reasoning? Qualitative examples could strengthen this claim.

2. hyperparameters setting
There are two sensitive hyperparameters: entropy-gated threshold and the minimum probability threshold.  It looks depends on models and tasks. Did you have any related experience?

---

> ### Author Response · Authors · 2025-11-21
> **Response to Reviewer rtNt (1/2)**
>
> We thank the reviewer for their time and the review. We are glad they recognized the novelty of integrating attribution into decoding and the strength of our visualizations. We would like to address the reviewer's concerns, starting with clarification about the paper's central claims and experimental setup.
>
> **[Clarification on tasks, datasets and contributions]**
> We would like to politely clarify that the reviewer's summary contains some inaccuracies that we kindly state here for reconsideration:
> * Instead of evaluating long summarization or reasoning, we evaluated our method on the following three tasks - **instruction following**, **closed-book factuality** and **open-book retrieval**.
> * Our primary application was instruction following, which we evaluated using the IHEval and SysBench benchmarks (Section 4, Table 1). The summary incorrectly states these measure factuality - they measure instruction adherence.
> * Our additional applications are closed-book factuality and open-book retrieval, evaluated on TriviaQA, Natural Questions, and HotPotQA (Section 5, Table 3 & 9).
>
> We hope this clarification helps contextualize our responses below.
>
> **[Limited novelty and relation to prior work]**
> Thank you for highlighting these additional related works. While these papers contribute valuable ideas to the field, they are not directly comparable to our approach for the following reasons:
> * **GeDi** [1] relies on additional task-specific fine-tuning (which can be computationally expensive) and requires a labeled dataset. In contrast, our method is entirely post-hoc: it requires no finetuning, no auxiliary data, and works out of the box.
> * **Unlocking Anticipatory Text Generation** [2] relies on external LLM judges to validate the sequence. A key concern is that the judge model itself can hallucinate or misinterpret constraints, so there are no strong guarantees about the guidance it provides. Our method instead uses attribution to trace the model’s internal representations, ensuring that the selected tokens genuinely follow from the instruction prompt and the model’s own computation. Moreover, we can define Region of Interest over internal components, which enables control beyond what natural language constraints can express.
>
> Please note that we **do not reweight token probabilities**. Once selected to the candidate set we don’t utilize any information about final distribution, the selection is performed solely based on the computed attributions. As such, rather than just reweighting token probabilities, we use explainable AI techniques to make LLM behavior more reliable.
>
> Once again, we appreciate the suggestions and we included them in our Related Work section in the revised version of the manuscript.
>
> **[Lack of human evaluation]**
> Our work follows an established paradigm in XAI research where attribution methods are used as **features for downstream tasks** rather than as explanations for human interpretation. Prior work has successfully applied XAI methods in fully automatic pipelines for tasks such as pruning [3], automatic model debugging [4] or continual learning support [5]. In these works attribution methods quantify component importance to enable algorithmic decisions, not to provide human-facing explanations. AGD operates in the same framework: we use attribution scores as **computable features** to guide token selection, similar to how attention patterns or gradient norms might be used algorithmically. With this context, we would like to address the specific evaluation concerns:
>
> * For factuality and in-context retrieval tasks we use standard datasets with ground-truth answers (TriviaQA, NQ, HotPotQA) and report recall metric. This is the established evaluation protocol - neither of our strong baselines CAD [6] nor DoLA [7] employ human evaluation for factuality assessment. When ground truth is available and quality can be objectively measured, human assessment is not necessarily needed.
> * In the instruction following task we share the reviewer’s concern about evaluating response quality beyond constraint adherence. Simply meeting instructions while ignoring the task would be useless - for instance, an empty response satisfies "don't use keyword X" instruction but has no utility. Therefore, apart from manual inspection of the samples, we implemented quality evaluation of generated responses with LLM-as-an-evaluator. The results are reported in Table 1 (QS) and the examples of qualitative evaluations are presented in Tables 2 & 6. This evaluation is done only on the task-response pair (instruction hidden), preventing the evaluator from utilizing constraint adherence in the quality evaluations. The whole procedure is described in detail in Section 4.1 and Appendix B (including prompts for LLM evaluator).
>
> Considering previous points, we believe that our evaluation approach provides comprehensive validation of AGD's effectiveness without requiring human evaluation studies.

---

> > ### Author Response · Authors · 2025-11-21
> > **Response to Reviewer rtNt (2/2)**
> >
> > **[There are two sensitive hyperparameters: entropy-gated threshold and the minimum probability threshold. It looks depends on models and tasks. Did you have any related experience?]**
> > We are happy to clarify this. We found these two hyperparameters to be robust and preserve high performance between different tasks and models.
> >
> > We selected a single set of hyperparameters - entropy-gated threshold $\tau$ as the 80th percentile of token-level entropy based on one model (Llama 3.1 8B) on one dataset (IHEval) and minimum probability threshold $\pi_{\min}$=0.05. As detailed in *Settings* paragraph in Section 4.1, to ensure fair comparison and demonstrate the generality of the method we then applied this exact same set of hyperparameters to all other models, tasks, and datasets without any further tuning. The consistent gains reported across all our experiments (Tables 1, 3, and 9) demonstrate that this single, non-tuned setting generalizes well, mitigating the concern that it requires careful hyperparameter setting for different models and tasks. We also provide sensitivity analysis of these hyperparameters in Section 4.3 (Figure 4).
> >
> > ---
> >
> > We appreciate the reviewer’s thoughtful feedback and recognize that some aspects of our manuscript, evaluation setup and quality assessment may not have been sufficiently clear. We hope that our clarifications help resolve these points and illustrate that AGD offers a novel, training-free approach that delivers consistent gains across diverse settings. We are happy to engage further and answer any questions that might arise. If the reviewer believes that we have sufficiently addressed their concerns, we would kindly ask to consider raising the score.
> >
> > ---
> >
> > **References:**
> > [1] GeDi: Generative Discriminator Guided Sequence Generation, Krause et al., EMNLP 2021
> > [2] Unlocking Anticipatory Text Generation: A Constrained Approach for Large Language Models Decoding, Tu et al., EMNLP 2024
> > [3] SNIP: Single-shot Network Pruning based on Connection Sensitivity, Lee et al., ICLR 2019
> > [4] Debugging tests for model explanations, Adebayo et al., Neurips 2020
> > [5] Explain to Not Forget: Defending Against Catastrophic Forgetting with XAI, Ede et al., CD-MAKE 2022
> > [6] Trusting Your Evidence: Hallucinate Less with Context-aware Decoding, Shi et al., NAACL 2024
> > [7] Dola: Decoding by contrasting layers improves factuality in large language models, Chuang et al., ICLR 2024

---

### Official Review · Reviewer_4ijK · 2025-10-29

**Soundness:** 2
**Presentation:** 2
**Contribution:** 2
**Rating:** 6
**Confidence:** 3

**Summary:**

- **Idea:** At each step, form a small candidate set (top-k with prob floor), compute attribution of each candidate to a user-defined Region of Interest (ROI) (e.g., instruction tokens, context embeddings, selected attention heads), and pick the token with the highest ROI attribution. No logits or activations are modified; it’s selection rather than intervention.
- **Variants:** Use LRP/AttnLRP (primary) or Input×Gradient for attribution; propose entropy gating to apply guidance only when the model is uncertain to curb quality loss and cost.
- **Results:** Improves instruction adherence on IHEval/SysBench and boosts factuality in closed-book (via “knowledge heads”) and open-book (via context embeddings or “in-context heads”); entropy gating trades off adherence vs quality and reduces backward passes.

**Strengths:**

- **Simple, model-agnostic control:** Decoding-time, fine-tuning-free; clear ROI abstraction spanning inputs or internals.
- **Practical quality/cost knob:** Entropy gating reduces intervention and keeps fluency while retaining most adherence gains.
- **Coverage:** Evaluated on instruction following (IHEval, SysBench) and factual QA (closed/open-book) across three small/mid models.

**Weaknesses:**

- **Positioning vs RAG/contrastive baselines is incomplete:** Open-book uses an ROI over context/heads, but no retrieval step; comparisons are to greedy/nucleus/CAD, not to RAG-style pipelines or “ROI-prompt only” ablations. This blurs how much gain comes from attribution versus simply highlighting the ROI.
- **Compute overhead and scalability:** AGD requires backward passes per candidate token when triggered; the paper reports entropy-gated reductions and average backward-pass counts, but lacks end-to-end wall-clock or token-cost comparisons against strong external baselines. Memory implications are not quantified.
- **Generalisation to stronger LLMs is unclear:** Results are on Llama-3.1-8B, Qwen-2.5-7B, Gemma-3-4B; it’s unknown whether gains persist or shrink on frontier small models (e.g., Qwen3, DeepSeek-R1-8B).

**Questions:**

1. **RAG vs AGD:** In open-book, could you compare against a standard RAG baseline (retrieve-then-decode) and a “ROI-prompt only” baseline that emphasises the ROI without attribution? This isolates attribution’s unique contribution.
2. **Cost & memory:** Please report wall-clock, token cost, and peak memory with and without entropy gating, and versus CAD/RAG on the same hardware. Also provide the average number of backward passes per token under typical thresholds.
3. **Stronger models:** Do gains hold for newer small models (e.g., Qwen3 or DeepSeek-R1-Qwen3-8B)? Any evidence of diminishing returns as base model adherence improves?
4. **Head selection robustness:** For “knowledge” or “in-context” heads, how stable are results across identification methods and datasets? Could you provide ablation of head sets and performance sensitivity?

---

> ### Author Response · Authors · 2025-11-20
> **Response to Reviewer 4ijK (1/3)**
>
> We sincerely thank the reviewer for their constructive feedback and detailed questions. We address each concern below with additional experiments and clarifications.
>
> **[RAG/contrastive baselines is incomplete, RAG vs AGD]**
> We appreciate the reviewer highlighting this important point. We clarify that our open-book experiments already use standard RAG-style prompting where context is provided directly in the prompt (Appendix A):
> > {{ context }} \n\nBased on this text, answer this question:\nQ: {{ question }}\nA:
>
> All baselines (greedy, nucleus, CAD) use this exact prompting structure. For TriviaQA and Natural Question datasets we provide a gold paragraph (containing answer) as a context, which makes them standard RAG baselines where the retrieval step is replaced by gold context provision. Since all methods receive the same prompt with the same context, AGD's gains over these baselines demonstrate the benefit of attribution-guided selection. In this setup, AGD is **complementary** to retrieval systems: retrieval finds the right information, while AGD ensures the model grounds its generation in that information.
>
> To address the concern about AGD's performance with imperfect retrieval, we conducted additional experiments - we simulated the RAG environment utilizing the HotPotQA distractor split, containing both relevant and irrelevant information. The context to the model is provided with 2 gold paragraphs (relevant) and 8 distractor paragraphs (noise), all shuffled randomly.
>
> **Table 1.** Results (Recall) with standard prompt (the one above) on HPQA distractor split:
> | **Method** | **Llama 3.1 (8B)** | **Qwen 2.5 (7B)** | **Gemma 3 (4B)** |
> |------------|-------------------|-------------------|------------------|
> | Greedy | 81.3 | 84.5 | 70.1 |
> | CAD | 83.7 | 85.0 | 64.1 |
> | AGD_LRPh | **87.9** | 86.4 | 70.6 |
> | AGD_LRPc | **87.9** | **89.5** | **70.9** |
>
> Since prompt sensitivity is a well-known issue in LLMs [1], we also tested a rephrased and reordered (question-first) prompt structure:
> > Question: {{ question }} \n\n Based on the text below, provide an answer:\n{{ context }}\nAnswer:.
>
> **Table 2.** Results (Recall) with reordered prompt on HPQA distractor split:
> | **Method** | **Llama 3.1 (8B)** | **Qwen 2.5 (7B)** | **Gemma 3 (4B)** |
> |------------|-------------------|-------------------|------------------|
> | Greedy | 82.2 | 83.5 | 62.7 |
> | CAD | 84.1 | 81.9 | 53.3 |
> | AGD_LRPh | **87.8** | 85.1 | **63.2** |
> | AGD_LRPc | 87.3 | **86.0** | 62.9 |
>
> As we can see, AGD maintains consistent advantages even with 80% noise in retrieved contexts. Moreover, gains persist with different prompt formulation, demonstrating robustness to prompt engineering. These results show that AGD's attribution mechanism provides value beyond oracle retrieval settings. It actively helps the model navigate noisy retrieval results by selecting tokens with stronger grounding in relevant portions of the provided context. These results further validate that the benefit of AGD comes from the decoding mechanism itself.

---

> > ### Author Response · Authors · 2025-11-20
> > **Response to Reviewer 4ijK (2/3)**
> >
> > **[Compute overhead and scalability]**
> > We thank the reviewer for pointing out this important topic. We provide comprehensive efficiency measurements below, conducted on NVIDIA A100 80GB GPUs using PyTorch 2.6.0 with bf16 precision. Metrics are averaged over 100 samples from the open-book TriviaQA dataset with maximum sequence length of 256 tokens.
> >
> > **Table 3.** Computational efficiency analysis. **Time**: average ms per token; **Cost**: slowdown relative to Greedy; **Memory**: peak GPU memory in MB; **Fwd/Bwd**: average number of forward/backward passes per token.
> > | **Method** | **Llama 3.1 (8B)** |  |  |  |  | **Qwen 2.5 (7B)** |  |  |  |  | **Gemma 3 (4B)** |  |  |  |  |
> > |------------|----------|--------|---------|-----|-----|----------|--------|---------|-----|-----|----------|--------|---------|-----|-----|
> > | | **Time (ms)** | **Cost×** | **Memory (MB)** | **Fwd** | **Bwd** | **Time (ms)** | **Cost×** | **Memory (MB)** | **Fwd** | **Bwd** | **Time (ms)** | **Cost×** | **Memory (MB)** | **Fwd** | **Bwd** |
> > | Greedy | 94.0 | 1.00× | 15520 | 1 | 0 | 85.5 | 1.00× | 14786 | 1 | 0 | 68.6 | 1.00× | 7578 | 1 | 0 |
> > | CAD | 124.9 | 1.32× | 16215 | 2 | 0 | 120.0 | 1.40× | 15566 | 2 | 0 | 134.8 | 1.97× | 9569 | 2 | 0 |
> > | AGD_LRPe | 150.7 | 1.60× | 23238 | 1 | 0.58 | 112.7 | 1.32× | 23628 | 1 | 0.23 | 92.1 | 1.34× | 14648 | 1 | 0.09 |
> > | AGD_LRP | 207.2 | 2.20× | 23842 | 1 | 1.06 | 161.6 | 1.89× | 23991 | 1 | 0.55 | 109.0 | 1.59× | 14739 | 1 | 0.34 |
> >
> > The entropy-gated version (AGD_LRPe ) provides a quite modest increase in compute ranging from 1.32 to 1.6x compared to greedy decoding, while non-gated requires 1.59-2.20x more time. This is comparable to the overhead of contrastive methods like CAD (1.32 - 1.97x), however our method requires more memory. This overhead comes, however, with performance gain, as shown by our results throughout the paper.
> >
> > Since we used local GPUs, reporting "token cost" (a common metric for commercial APIs) is not relevant, and wall-clock time is the appropriate hardware-specific metric.
> >
> > **[Generalisation to stronger LLMs is unclear]**
> > We thank the reviewer for this important question about generalization to newer models. We have conducted preliminary experiments on Qwen3-4B-Instruct-2507, a newer model with improved base capabilities.
> >
> > **Table 4.** Prompt-level accuracy (Instruction-level accuracy) on IHEval dataset
> > | **Method** | **PLA (ILA)** |
> > |------------|---------------|
> > | Greedy | 83.0 (88.6) |
> > | AGD_LRPe | 84.8 (89.7) |
> > | AGD_LRP | **86.3 (91.0)** |
> >
> > **Table 5.** Results on TQA dataset
> > | **Method** | **Recall** |
> > |------------|---------|
> > | Greedy | 91.7 |
> > | AGD_LRPc | **92.4** |
> >
> > These results suggest that AGD continues to provide gains on newer models, though the magnitude is a bit smaller than for Llama 3.1 (8B) and Qwen 2.5 (7B), however comparable to Gemma 3 (4B).
> >
> > We acknowledge that these are early results and, due to computational constraints we have not yet completed comprehensive evaluation across all tasks and baselines for Qwen 3 4B, nor more recent models of bigger size. In the camera-ready version, we will provide a more comprehensive evaluation for the newer models.

---

> > > ### Author Response · Authors · 2025-11-21
> > > **Response to Reviewer 4ijK (3/3)**
> > >
> > > **[Head selection robustness]**
> > > We conducted ablation studies on the number of heads selected for AGD on TriviaQA with Llama 3.1 8B. Heads are sorted by their "knowledge" or "in-context" capabilities (using the methodology described in Appendix D) and selected from the top.
> > >
> > > **Table 6.** Results (Recall) for ablation study of head count
> > > | **#heads** | **Factuality** | **In-Context** |
> > > |------------|----------------|----------------|
> > > | 10 | 81.8 | 90.3 |
> > > | 50 | 81.4 | **91.4** |
> > > | 100 | **82.4** | 91.0 |
> > > | 200 | **82.4** | 90.9 |
> > >
> > > The performance is relatively stable across head counts from 50-200. Very small head sets (N=10) underperform - likely we consider too little number of heads and coverage insufficiently the relevant information pathways. We observe no significant degradation with larger head sets, suggesting that the sorting effectively prioritizes relevant heads. We suspected that too big of a number could introduce noise, but it doesn’t seem to be the case. Overall, our choice of 100 heads ($\sim$10\% of total heads) represents a reasonable middle ground.
> > >
> > > Our head identification follows the established methodology of [2], which uses counterfactual attribution on the NQ-Swap dataset. While exploring alternative identification methods (e.g., perturbation-based) is interesting future work, it’s out of the score of this paper. The stability across different head counts suggests that the core AGD mechanism is not overly sensitive to precise head selection, as long as generally relevant heads are targeted.
> > >
> > > ---
> > >
> > > We hope these additional experiments and clarifications adequately address the reviewer's concerns. We believe the new results, particularly the noisy retrieval experiments with different formatting, comprehensive efficiency analysis, and head selection ablation study strengthen the paper's empirical foundation and robustness. We updated the manuscript to incorporate the feedback and additional results (Section 5, Appendix D, E and G). If the reviewer finds that our clarifications and additional results meet their expectations, we would kindly ask to consider raising their score. We are happy to discuss any further questions the reviewer might have.
> > >
> > > ---
> > >
> > > **References:**
> > > [1] Quantifying Language Models' Sensitivity to Spurious Features in Prompt Design or: How I learned to start worrying about prompt formatting, Sclar et al., ICLR 2024
> > >
> > > [2] The Atlas of In-Context Learning: How Attention Heads Shape In-Context Retrieval Augmentation, Kahardipraja et al., NeurIPS 2025

---

### Official Review · Reviewer_neLF · 2025-11-01

**Soundness:** 3
**Presentation:** 3
**Contribution:** 3
**Rating:** 6
**Confidence:** 3

**Summary:**

This paper presents Attribution-Guided Decoding (AGD), a fine-tuning-free decoding strategy to enhance the reliability of LLMs. AGD is based on the selection of tokens from a set of high-probability candidates. Selection is based on the maximization of the attribution score of a candidate token to a user-defined "Region of Interest" (ROI). This ROI can be flexibly defined over parts of the input prompt or over internal model components. The authors show the effectiveness of AGD in improving instruction following, increasing factual accuracy in closed-book question answering by avoiding hallucinations, and enhancing grounding in open-book (in-context) retrieval tasks. Regarding the computational overhead of calculating attributions at each step, an adaptive variant is also proposed, entropy-gated AGD-e, which applies guidance only when the model output distribution shows high uncertainty, thus preserving the quality of the output while saving cost.

**Strengths:**

1. AGD repurposes passive attribution methods for active generation. Unlike "interventionist" methods that alter activations, this "selectionist" approach chooses among the model's high-probability outputs, preserving text quality while making generation interpretable through token-rationale relationships.

2. The ROI framework applies broadly to diverse problems. AGD improves instruction adherence (using instruction text as ROI), factual recall (targeting "parametric knowledge heads"), and in-context grounding (using context embeddings as ROI).

3. The authors tested across multiple models (Llama 3.1, Qwen 2.5, Gemma 3) and tasks with strong baselines like CAD and DoLA. The analysis includes hyperparameter ablations and case studies visualizing attribution scores to explain why the method works.

**Weaknesses:**

1. AGD's "selectionist" design limits it to tokens already proposed by the base model with high probability. If the desired token isn't in the candidate set, AGD cannot generate it, making success dependent on the base model's capabilities.

2.  The method seems to be computationally expensive, requiring backward passes for each candidate token to compute attribution scores. While entropy-gating helps, this overhead could limit its practice. In the meantime, we need to predefine the threshold for the entropy, which can not always be reliable.

3. While ROI flexibility is a strength, it requires users to define relevant ROIs for each task. This is straightforward for instruction following but non-trivial for tasks like factuality improvement, which requires identifying specialized components like "knowledge heads."

4. Many baselines are missing such as DoLA.

**Questions:**

see weakness

---

> ### Author Response · Authors · 2025-11-20
> **Response to Reviewer neLF**
>
> We thank the reviewer for the detailed review and thoughtful feedback. We are glad they recognized the novelty of our "selectionist" approach, the broad applicability of the ROI framework, and the strength of our experimental setup. We appreciate the constructive feedback and will address the raised concerns below:
>
> **[If the desired token isn't in the candidate set, AGD cannot generate it]**
> We agree with the reviewer that this is an inherent and important characteristic of our "selectionist" design. This is by design - we cannot generate such token, however it ensures that AGD preserves the base model's fluency and coherence by selecting from plausible outputs (similarly to nucleus sampling for example) rather than forcing it into out-of-distribution states (a common issue with methods that directly manipulate activations).
> The reviewer's point is correct - AGD is designed to guide a model, not to fix one. This also means that as base models continue to improve and propose better candidate sets, we suspect the performance of AGD may scale alongside them, making it a robust guidance technique.
>
> **[method seems to be computationally expensive]**
> We agree that AGD incurs an additional overhead due to the backward passes, which we mitigate to some extent by introducing an entropy-gated version. We performed additional computational cost analysis, the results can be found in Appendix E. Entropy-gated AGD achieves only 1.32-1.60x (across different models) slowdown compared to the greedy decoding while maintaining substantial performance gains. Ultimately, we view this as a clear **trade-off**: AGD exchanges compute for significant gains in reliability, control, and task-specific performance, which is a worthwhile trade in many high-stakes applications.
>
> **[In the meantime, we need to predefine the threshold for the entropy, which can not always be reliable.]**
> Regarding the entropy threshold, we found it to be more robust than the reviewer might expect. As noted in the paper (Settings paragraph in Section 4.1), in our experiments we used a single percentile threshold (the 80th) derived from one model (Llama 3.1 8B) on a single dataset. This same threshold was then applied, without re-tuning, to all other models and datasets. The results presented in our paper (e.g., in Table 1) demonstrate that this threshold transfers well, mitigating the cost without requiring task-specific hyperparameter tuning. The sensitivity of the performance wrt. to this threshold is illustrated in Figure 4a.
>
> **[ROI flexibility]**
> This is a valid point, which we also acknowledged in our limitations (Conclusion, line ~500). However, we would like to emphasize that for many common use cases, the ROI is straightforward to define:
>
> * For instruction following, the ROI is simply the instruction text.
> * For open-book grounding (RAG), it is the provided context.
>
> The more complex ROI definition (e.g., "knowledge heads") is presented as an advanced application to demonstrate the versatility of the method. We are excited by this flexibility, as it opens up new research directions into identifying and leveraging functional components (e.g., circuits) within LLMs.
>
> **[Missing baselines, such as DoLA]**
> If the reviewer meant that DoLA is missing, we would like to point out that, in fact, **DoLA** is included as a strong baseline for the closed-book factuality task (as the reviewer correctly pointed out in Point 3 of the Strengths section). The results for DoLA are presented in Table 3 (top section) and Table 9, where AGD (specifically AGD_LRPh) outperforms it on almost all datasets across different models. We also included **CAD (Context-aware Decoding)**, another strong, relevant baseline, for the open-book task in Table 3 (bottom section) and Table 9. We believe these comparisons, along with other standard decoding methods (greedy, nucleus) provide a strong and relevant benchmark for AGD's performance.
>
> ---
> We thank the reviewer again for their feedback and hope we have convincingly addressed their concerns. If so, we kindly ask the reviewer to consider raising their score. In any case, we are happy to discuss any remaining concerns the reviewer might have.

---

### Author Response · Authors · 2025-12-03
**Final Remarks**

We thank the reviewers for their feedback and would like to summarize the key points from the rebuttal.

**[Acknowledged strengths]**
We would like to briefly restate the core contributions of our paper and thank reviewers for acknowledging them in their reviews:
* **Novel "selectionist" paradigm for controlled generation** - Attribution-Guided Decoding (AGD) selects among the model's high-probability candidates using attribution scores rather than manipulating activations or final logits, preserving fluency while enabling control. To our knowledge, this is the first work to repurpose attribution methods for active generation during decoding. (`neLF`, `4ijK`)
* **Flexible Region of Interest (ROI) framework** - By flexible definition of ROI over different components one can apply our method to different tasks (e.g. part of the prompt for instruction following or identified attention heads for factuality/in-context retrieval). This also opens possibilities for leveraging identified functional components within LLMs (e.g. circuits) in the future work. (`neLF`, `4ijK`)
* **Practical entropy-gating mechanism** - Adaptive variant applies guidance only when the model is uncertain, providing an alternative that reduces computational overhead while preserving most gains. (`neLF`, `4ijK`, `rtNt`)
* **Comprehensive evaluation** - Tested across three tasks (instruction following, closed-book factuality, in-context retrieval), three model families (Llama 3.1, Qwen 2.5, Gemma 3), multiple benchmarks (IHEval, SysBench, TriviaQA, Natural Questions, HotPotQA) with relevant and strong baselines (CAD, DoLA), including hyperparameter ablations and case studies. (`neLF`, `4ijK`, `rtNt`)

**[Concerns addressed with new experiments]**
We conducted significant additional work during rebuttal to address reviewers concerns and updated the manuscript:
* **RAG/retrieval robustness** (`4ijK`) - we performed in-context retrieval experiments with 80% distractor noise in the provided context and found that the gains persist. AGD improves +6.6 points over greedy despite noisy retrieval. (Section 5, Appendix F, G)
* **Prompt sensitivity** (`4ijK`) - we tested two distinct prompt formulations and reported consistent advantages across both variants. (Appendix G)
* **Computational cost** (`neLF`, `4ijK`) - we performed efficiency analysis - it shows that, while non-gated version introduce more significant computational overhead, the entropy-gated version is comparable to contrastive decoding method (CAD). (Appendix E)
* **Generalization to newer models** (`4ijK`) - we performed preliminary evaluation on Qwen3-4B-Instruct-2507 and reported persistent gains with our method (86.3% vs 83.0% PLA on IHEval).
* **Head selection sensitivity** (`4ijK`) - we studied ablations across 10-200 heads and showed stable performance results in the range of 50-200 heads. (Appendix D)

We believe these revisions considerably strengthen the paper's empirical foundation.

---

### Meta-Review · Area_Chair_eUpA · 2026-01-08

**Summary:**

This paper proposes a method to guide the generation process using attribution. Given user-defined regions of interest (ROIs), the proposed method selects tokens by maximizing attribution scores, thereby steering the generation. Experimental results demonstrate the effectiveness of the method in improving instruction following and reducing hallucinations.

Overall, reviewers acknowledge the contribution of this work, and the AC agrees with this assessment. Most of the concerns are addressed by additional results during the rebuttal. Some limitations were raised by reviewers, including high computational cost, reliance on user-defined ROIs, and the limitation of the candidate set. AC agrees with these limitations but believe they are acceptable or can be explored in future work. AC also believes that incorporating some level of human evaluation in the final version could further strengthen the paper, as suggested by the reviewers.

**Reviewer Concerns:**

- (Fully addressed) Reviewers asked for a comparison with RAG. The authors provided additional results.
- (Fully addressed) Reviewers asked for comparisons with more LLMs. The authors provided additional results.
- (Fully addressed) Other clarification questions were properly addressed.
- (Not addressed) Reviewers believe that human evaluation is needed.
- Reviewers noted several limitations, including high computational cost, requirement on user-defined ROIs, and that the method cannot generate tokens beyond a candidate set. The authors provided some explanation.

**Reviewer Scores:**

I believe all reviewers will maintain their scores.

---

### Decision · Program_Chairs · 2026-01-26

Accept (Poster)